


# Meteorological, Impact and Climate perspectives of the 29 June 2017 Heavy Precipitation Event in the Berlin Metropolitan Area

Alberto Caldas-Alvarez[1], Markus Augenstein[1], Georgy Ayzel[2], Klemens Barfus[3], Ribu Cherian[4], Lisa Dillenardt[2], Felix Fauer[5], Hendrik Feldmann[1], Maik Heistermann[2], Alexia Karwat[6], Frank Kaspar[7], Heidi Kreibich[8], Etor Emanuel Lucio-Eceiza[5,9], Edmund P. Meredith[5], Susanna Mohr[1,10], Deborah Niermann[7], Stephan Pfahl[5], Florian Ruff[5], Henning W. Rust[5], Lukas Schoppa[2,8], Thomas Schwitalla[11], Stella Steidl[7], Annegret H. Thieken[2], Jordis S. Tradowsky[12,13], Volker Wulfmeyer[11], and Johannes Quaas[4]

[1]Institute of Meteorology and Climate Research (IMK-TRO), Karlsruhe Institute of Technology (KIT), Karlsruhe, Germany
[2]Universität Potsdam, Institute of Environmental Science and Geography, Karl-Liebknecht-Str. 24-25, 14476 Potsdam, Germany
[3]Technische Universität Dresden, Institute of Hydrology and Meteorology, Pienner Straße 23, 01737 Tharandt, Germany
[4]Institute for Meteorology, Universität Leipzig, Leipzig, Germany
[5]Freie Universität Berlin, Institute of Meteorology, Carl-Heinrich-Becker-Weg 6-10, 12165 Berlin, Germany
[6]Universität Hamburg, Meteorological Institute, Grindelberg 5, 20144 Hamburg, Germany
[7]Deutscher Wetterdienst, Frankfurter Straße 135, 63067 Offenbach am Main
[8]Section Hydrology, GFZ German Research Centre for Geosciences, Telegrafenberg, 14473 Potsdam, Germany
[9]Deutsches Klimarechenzentrum, Bundesstraße 45a, 20146 Hamburg, Germany
[10]Center for Disaster Management and Risk Reduction Technology (CEDIM), Karlsruhe Institute of Technology (KIT), Karlsruhe, Germany
[11]Institute of Physics and Meteorology, University of Hohenheim, Garbenstrasse 30, 70599 Stuttgart, Germany
[12]Deutscher Wetterdienst, Regionales Klimabüro Potsdam, Güterfelder Damm 87-91 14532 Stahnsdorf, Germany
[13]Bodeker Scientific, 42 Russell Street, Alexandra 9391, New Zealand

**Correspondence:** Alberto Caldas-Alvarez (alberto.caldas-alvarez@kit.edu) and Johannes Quaas (j.quaas@uni-leipzig.de)

**Abstract.** Extreme precipitation is a weather phenomenon with tremendous damaging potential for property and human life. As the intensity and frequency of such events is projected to increase in a warming climate, there is an urgent need to advance the existing knowledge on extreme precipitation processes, statistics and impacts across scales. To this end, a working group within the German-based project ClimXtreme, has been established to carry out multidisciplinary analyses of high-impact events. In

5    this work, we provide a comprehensive assessment of a selected case, affecting the Berlin metropolitan region (Germany) on 29 June 2017, from the meteorological, impacts and climate perspectives, additionally estimating the contribution of climate change to its extremeness. Our analysis shows that this event occurred under the influence of a mid-tropospheric trough over western Europe and two short-wave surface lows over Britain and Poland, inducing relevant low-level wind convergence along the German-Polish border. Several thousand convective cells were triggered in the early morning of 29 June, displacing

10    northwards slowly under the influence of a weak tropospheric flow ($10\,\mathrm{m\,s^{-1}}$ at 500 hPa). A very moist and warm southwesterly flow was present south of the cyclone over Poland, in the presence of moderate Convective Available Potential Energy (CAPE). We identified the soil in the Alpine-Slovenian region as the major moisture source for this case (63 % of identified sources). Maximum precipitation amounted to $196\,\mathrm{mm\,d^{-1}}$, causing the largest insured losses due to a heavy precipitation event in the





period 2002 to 2017 (€60 Mill.) over the area. A household-level survey revealed that the inundation duration was 4 to 12
times larger than other surveyed events in Germany in 2005, 2010 and 2014. The climate analysis showed return periods of
over 100 years for daily aggregated precipitation, and up to 100 years and 10 years for 8 h and 1 h aggregations, respectively.
The event was the $29^{th}$ most extreme event in the 1951-2021 climatology in terms of severity and the second with respect to the
number of convective cells triggered from 2001 to 2020 over Germany. The conditional attribution demonstrated that warming
since the pre-industrial era caused a small, but significant increase of 4 % in total precipitation and 10 % for extreme intensities.
The aerosol sensitivity experiments showed that increased anthropogenic aerosols induce larger cloud cover and probability
of extreme precipitation ($> 150 \, \mathrm{mm \, d^{-1}}$). Our analysis allowed relating interconnected aspects of extreme precipitation. For
instance, the link between the unique meteorological conditions of this case and its climate extremeness, or the extent to which
this is attributable to already-observed anthropogenic climate change.

## 1 Introduction

According to the World Economic Forum Global Risks Perception Survey (Forum, 2020), extreme weather is the number one
risk by likelihood and among the top four risks by impact. One of the most impactful weather types is extreme precipitation,
which yearly causes local ecosystems and urban areas to suffer important damages and casualties. The probability of occurrence
and magnitude of this extreme weather is projected to increase in northern and central Europe in a warming climate, as assessed
by the 6th Assessment Report of the Intergovernmental Panel on Climate Change (Douville et al., 2021).

The interaction of processes across scales hampers our comprehension and prediction of Heavy Precipitation Events (HPEs).
Extreme precipitation will occur only under favourable synoptic-scale conditions (Brieber and Hoy, 2018) with sufficient
moisture transport (Davolio et al., 2020; Caldas-Alvarez et al., 2021), and atmospheric instability (Khodayar et al., 2021),
fostered by propitious phases of climate modes (Ehmele et al., 2020). This complexity is further amplified if the impacts of
heavy precipitation are to be addressed. In addition to the intensity of the hazard, the impact of an event depends on the exposure
and vulnerability of the affected area (Alfieri et al., 2015). This is why multidisciplinary, forensic analysis is a powerful means
to deal with the complex interactions underlying an HPE and its impacts. Forensic analysis consists of addressing different
aspects of heavy precipitation jointly, so that the interconnections between findings from different disciplines can be identified.
For instance, Bronstert et al. (2018) investigated the Braunsbach flood in 2016, a flash flood event in a sparsely observed area,
analysing the damages in the built-up area as well as its geomorphological impacts. Kunz et al. (2013) analysed Hurricane
Sandy in 2012, combining an in-depth assessment of its impacts with the usage of information from social networks for
event reconstruction. Gochis et al. (2015) and Milrad et al. (2015) presented detailed post-event analyses using measurements,
operational data products and application of models especially suited for widespread events in well-observed areas for the
Colorado and Alberta floods (2013), respectively. Finally, other studies (e.g., Eden et al., 2016) have complemented forensic
studies with climate change attribution experiments.

The favourable synoptic conditions for HPE development in central Europe have been assessed in previous literature. Werner
and Gerstengarbe (2010), using weather pattern classification, concluded that summer HPEs over Central Europe are often





caused by three synoptic situations, a Trough over Central Europe (Tr), low pressure over Central Europe (TM) and a Trough over West Europe (TrW), see also Wulfmeyer et al. (2011). Brieber and Hoy (2018) found the highest probabilities for heavy precipitation events in central Germany when a TrW pattern is present, favouring the development of small-scale disturbances
such as Vb-like cyclones or heat lows and prefrontal convergence zones. Depending on the location of the small-scale disturbances, warm and moist air masses are transported from southern Europe towards Germany. When temperatures are already very high, the resulting increased CAPE leads to localized extreme precipitation (Bronstert et al., 2018).

A full understanding of the meteorological drivers and (small-scale) physical processes of extreme precipitation is often restricted by insufficient high-resolution observations (Wulfmeyer et al., 2008, 2020). Numerical models offer a robust tool to
simulate extreme precipitation events with fine-scale spatio-temporal detail. Furthermore, numerical models can also be used to create climate time series long enough to capture multi-decadal variability and numerous extreme events (Ehmele et al., 2020; Pichelli et al., 2021). Over the last decade, increased computing power has seen the growing use of kilometre-scale "storm-resolving" or convection-permitting models (CPMs; Berg et al., 2012; Barthlott and Hoose, 2015; Schwitalla et al., 2020; Stevens et al., 2020; Lucas-Picher et al., 2021), in which spatial resolution is sufficiently high ($\Delta x < 3$ km) to explicitly
simulate deep convection. CPMs have thus shown added value for the simulation of sub-daily extreme precipitation intensities, their spatial extent and duration, as well as their diurnal cycles (Kendon et al., 2012; Warrach-Sagi et al., 2013; Fosser et al., 2014; Stevens et al., 2020; Meredith et al., 2021). This added value offers important utility for climate-change attribution studies since CPMs give a better representation of convective extremes and are thus more reliable for detecting their climate-change response (Prein et al., 2013, 2015; Ban et al., 2021).

The impacts of extreme precipitation and the resulting pluvial flooding on society can include the loss of life, physical damage to assets such as buildings or infrastructure, as well as intangible consequences such as health impacts or traffic disruptions (Merz et al., 2010; Rözer et al., 2016). The impacts depend on the hazard intensity, the exposed assets and their vulnerability (Kron, 2005). The hazard level itself is a function of meteorological factors, e.g. precipitation intensity or affected area. The exposure and vulnerability depend on aspects such as the inundation depth, the topography, the degree of sealing of
surfaces, or the sewer system capacity (Kron, 2005; Smith et al., 2015). Hence, whether a flood causes damage depends not only on the development of a meteorological situation, but also on the number and types of exposed assets and their vulnerability.

To analyse the associated impacts, not only meteorological data is crucial, but also targeted information collected during or after an event, e.g. damage to buildings and contents. Such data can be collected, for example, through surveys of affected households or through assessments by loss surveyors during insurance claim validation (Spekkers et al., 2014, 2017; Van
Ootegem et al., 2015, 2018; Rözer et al., 2019). However, when analysing damage records, it is important to consider that results depend on how the affected society understands, records, and remembers those impacts (Kuhlicke et al., 2020). The case under study here, the 29 June 2017 event in the metropolitan Berlin area (Germany), has been investigated before based on survey data of affected households in Berlin-Brandenburg. Berghäuser et al. (2021) and Dillenardt et al. (2021) addressed the tangible and intangible consequences of extreme precipitation and associated pluvial flooding for households for this event,
respectively, but a focused analysis comparing its impacts to those of similar HPEs in the climatology is still lacking.



From the climate perspective, information on how the frequency of HPEs has changed in the recent climate is demanded by interested stakeholders. This can be provided through estimation of probabilities of exceedance or return periods for specific events. Generalized Extreme Value (GEV) models can be fitted to a climate observational precipitation dataset to derive this information (Wilks, 2006). Previous studies have provided estimations of return periods for similar events, finding extreme

values of over 200 years for the Seine river flooding (France) in 2016 (Philip et al., 2018), the Braunsbach flooding (southern Germany) in 2016 (Piper et al., 2016) and the three-week flooding in Germany in 2018 (Mohr et al., 2020). The flooding in July 2021 in the Ahr, Erft and Meuse rivers, was analysed by Kreienkamp et al. (2021), who found that an event of similar meteorological characteristics can be expected in the present climate in Central Europe once every 400 years.

Also at the climate scale, extreme event attribution has proven useful to estimate how the severity and/or likelihood of an

event has been affected by anthropogenic influences (Allen, 2003; Stott et al., 2004; Otto, 2017). Anthropogenic influence typically refers to climate change, but could also include, e.g., land-use changes (Sebastian et al., 2019) or changes in atmospheric pollutants (Liu et al., 2020). To this end there are two mainly-used approaches. The first approach is probabilistic event attribution, which consists of simulating how the dynamics of the climate system evolve under climate change. It can therefore be used to detect significant changes in the severity and frequency of extreme events. This technique, however, is applicable

to model data with a relatively coarse resolution and is, therefore, best suited for attribution studies of large-scale events. The second approach is conditional event attribution (Trenberth et al., 2015), which evaluates to what extent observed climate change may have impacted the magnitude of an event. The attribution is thus conditional on the presence of a given dynamical situation, and implies that thermodynamic changes due to climate change have been demonstrated. However, a limitation of this approach is that changes in the probability of the event's underlying dynamical situation cannot be determined.

The research presented here has been carried out in the framework of the Germany-based project Climate change and eXtreme events (ClimXtreme[1]), which brings together governmental and research institutions in the fields of atmospheric physics, statistics, impact studies and computing. ClimXtreme deals with the influence of climate change on atmospheric extremes. To facilitate reproducibility and cooperation between project members, ClimXtreme uses the ClimXtreme Central Evaluation System (XCES), a scientific software infrastructure (Kadow et al., 2021) that allows centralized consultation and

analysis of all the data within the project.

The aim of this study is to provide a comprehensive assessment of the 29 June 2017 HPE in the area of Berlin (Germany) from the meteorological, impacts and climate perspective, additionally estimating the contribution of climate change to its extremeness. The paper is structured as follows: in Section 2 we present the datasets and methods used for analysis. In Section 3 we describe our results from the multidisciplinary analyses. Finally, in Section 4 we introduce our conclusions, findings and

outlook.

---

[1] https://www.ClimXtreme.net/index.php/en/





## 2 Data and methods

### 2.1 Atmospheric observations and reanalyses

Our analysis is based on precipitation, lighting and survey observations. Where observations were not available, we utilize
climate reanalyses.

**2.1.1 Precipitation**

**REGionalisierte NIEederschlaege (REGNIE)**

REGNIE is a gridded data set of 24-hour totals (from 06 UTC to 06 UTC) based on approximately 2,000 rain gauges distributed
across Germany. A post-processing is applied to station data for regridding to a $1 \times 1 \, \text{km}^2$ mesh taking into account elevation,
exposure and climatology, avoiding smoothing observed precipitation extremes (Rauthe et al., 2013; Hu and Franzke, 2020).

The data are provided by the German Weather Service (DWD) from 1951 for all of Germany (for the former West Germany,
the daily values are available since 1931). The long-term availability of REGNIE is its main advantage for climate studies.

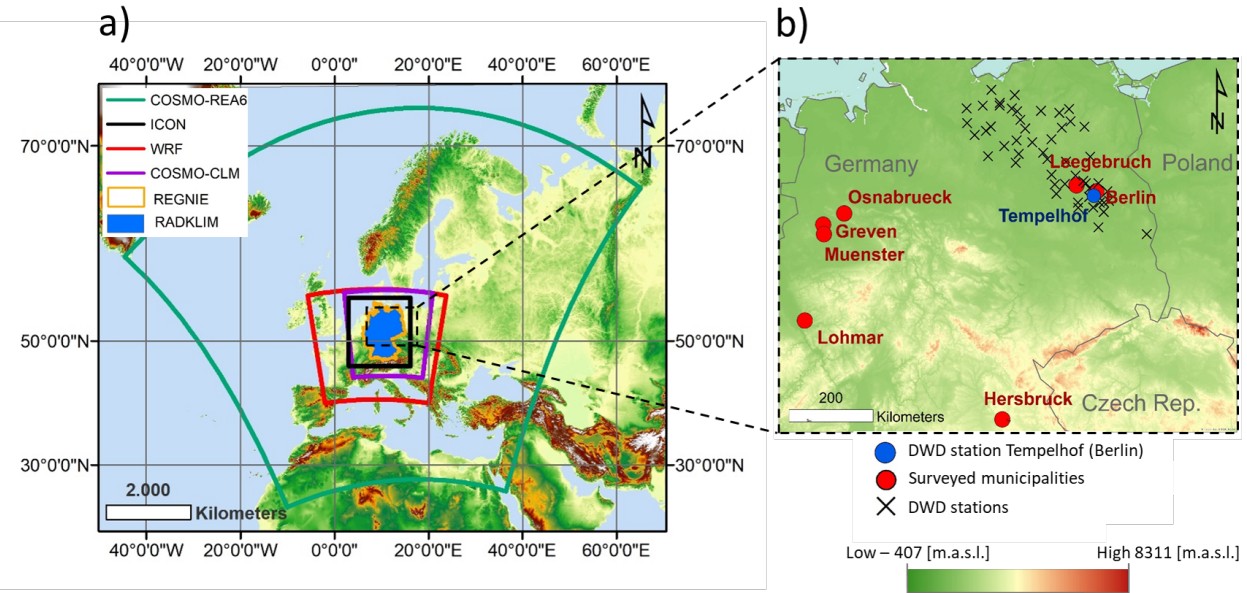

**Figure 1.** a) Spatial extent of the used data sets REGNIE, RADKLIM, COSMO-REA6, WRF-1.5km, ICON-625m and COSMO-CLM-2.8.
b) Close-up view of Berlin and surroundings, as well as the DWD stations used for validation (crosses), the location of surveyed events from
Table 1 (red dots) and the Berlin-Tempelhof station analysed in Figure 10 (blue dot), Basemap: Global Land One-kilometer Base Elevation
(GLOBE) (Hastings et al., 1999).


**RADar KLIMatologie (RADKLIM)**

RADKLIM is a precipitation climate data set derived from the C-band radar network (17 radar sites) operated by DWD.
The data set comprises two products; the gauge-adjusted one-hour precipitation sum (RW) and the quasi gauge-adjusted five-minute precipitation rate (YW), with $1 \times 1$ km$^2$ resolution. The data are post-processed using the Radar-Online-Aneichung (RADOLAN) method, correcting existing artifacts by adjusting precipitation sums from the radar with precipitation measurements from rain gauge stations (Bartels et al., 2004). Due to the dense spatial and temporal resolution, RADKLIM detects short-term, convective extreme intensities frequently missed by station data (Lengfeld et al., 2020; Winterrath et al., 2017). It
is available between 2001 and 2020.

**CAtalogue of Radar-based heavy Rainfall-Events (CatRaRE)**

The RADKLIM data set at 1 h and $1 \times 1$ km$^2$ km resolution (RW) was used as the basis to derive an HPE catalogue for Germany (CatRaRE) for the period 2001 to 2020. Here we use Version 2017.002 (Lengfeld et al., 2021a, b), including precipitation sums
with 11 different accumulation periods (1, 2, 3, 4, 6, 9, 12, 18, 24, 48 and 72 hours). For each duration and time step, extreme events are detected based on the DWD Warning Level (WL) 3 for severe weather, or if they have a return period of 5 years. The duration in hours, the affected area in km$^2$, the location (county and community), the maximum and mean precipitation amount in mm as well as affected residents in the event area and further meta information is included.

### 2.1.2 Lightning data

The lightning activity in and around Berlin is investigated with data from the ground-based low-frequency lightning detection system operated by Siemens as part of the EUropean Cooperation for LIghtning Detection (EUCLID) network, which covers the whole European continent (Drüe et al., 2007; Schulz et al., 2016; Poelman et al., 2016). Cloud-to-ground strokes are used to illustrate the temporal development of convective activity during the extreme event, similar to Piper and Kunz (2017) and Wilhelm et al. (2021). The spatial resolution of EUCLID has been improved to less than 90 m in the year 2016 due to algorithm
optimizations (Schulz et al., 2016).

### 2.1.3 Reanalyses

**ERA5**

ERA5 data are used to identify and track the low pressure systems, find moisture sources and force numerical simulations. This is the fifth generation of the European Center for Medium-Range Weather Forecasts' (ECMWF) atmospheric reanalysis of the
global climate (Hersbach et al., 2020). It has a spatial resolution of $31 \times 31$ km$^2$ and is available from 1950 to present at hourly resolution. ERA5 is based on the Integrated Forecasting System (IFS) and uses a 4D-Var assimilation scheme, assimilating different observation types. ERA5 has demonstrated a good performance in representing heavy precipitation (Keller and Wahl, 2021).





## 2.2 Validation of precipitation data sets

To ensure consistency of our results, we provide a quantitative validation of the used precipitation products and of ERA5. REGNIE, RADKLIM and ERA5 are compared pointwise (by selecting the nearest neighbour grid cell) to observations made at 53 DWD stations, located in the area (Fig. 1b). Additionally, the regional reanalysis COnsortium for Small-Scale MOdeling - Reanalysis 6 km (COSMO-REA6) is compared (Bollmeyer et al., 2015) to provide a reference for ERA5 with another reanalysis product. COSMO-REA6 was developed by the University of Bonn and the DWD within the Hans-Ertel-Centre for

Weather Research (Simmer et al., 2016). It has about $6 \times 6 \ \mathrm{km}^2$ horizontal resolution and accumulated precipitation is provided hourly (Bollmeyer et al., 2015).

Daily precipitation measurements from the DWD high-density network were used as reference. The data are quality controlled, but a continuous homogenization is not applied. Hence, the data could be subject to partial inhomogeneities, such as station relocations or changes in the instrumentation (Kaspar et al., 2013). The DWD high-density network has a high accuracy

and resolution (Kaspar et al., 2013) and therefore we select it for validation. Nonetheless, this dataset is used for deriving the REGNIE gridded product and for adjusting RADKLIM which makes these data sets dependent. We concede this dependency to profit from the best rain gauges product available in the region.

In the validation, different scores are computed based on daily precipitation sums over the years 2001 to 2018, which is the longest period for which all data sets are available. We compute absolute frequencies (Fig. 2.a), the Symmetric Extremal

Dependency Index (SEDI; Fig. 2.b) and the frequency bias (Fig. 2.c). The absolute frequencies show the number of days with a specific amount of precipitation for each data set. The SEDI estimates the dependency between an event in the given data set and the reference observations (Ferro and Stephenson, 2011). Finally, the frequency bias describes the ratio between the number of events in the data sets and the reference observations (Hogan et al., 2009). The evaluations are made for three precipitation intervals based on the WLs used operationally by the DWD, namely WL2 (> 30 mm), WL3 (> 50 mm) and

WL4 (> 80 mm). The observed 29 June event falls into the WL4 category.

REGNIE fits best the observations for absolute frequency (37 events in REGNIE, 34 in the DWD stations at WL4), followed by RADKLIM (24 events in WL4) and ERA5 (Fig.2.a). The latter only has 12 events (WL4), which is probably due to its coarse resolution and the inability to explicitly resolve structures smaller than $25 \times 25 \ \mathrm{km}^2$. This impact of model resolution is further evidenced by the performance of the second reanalysis COSMO-REA6 with 36 events in WL4. Despite both reanalyses

using deep convection parameterizations, COSMO-REA6 is able to represent larger precipitation intensities and hence more HPEs. Regarding SEDI, REGNIE again compares best with station observations, also followed by RADKLIM (Fig. 2b). In contrast, the coarser reanalysis products show lower, but acceptable values, especially for WL4. Finally, for frequency bias, REGNIE shows no deviations, RADKLIM has a negative bias, i.e. less events detected, and ERA5 performs poorly for all WLs.

We conclude that REGNIE and RADKLIM are well suited for process-based and statistical analyses of extreme precipitation at the meteorological and climate scales and that ERA5 performed badly in the comparison against the DWD stations. However, the good results of ERA5 in SEDI for extreme events (WL4; Fig. 2.b) and the accuracy shown by previous studies for large-





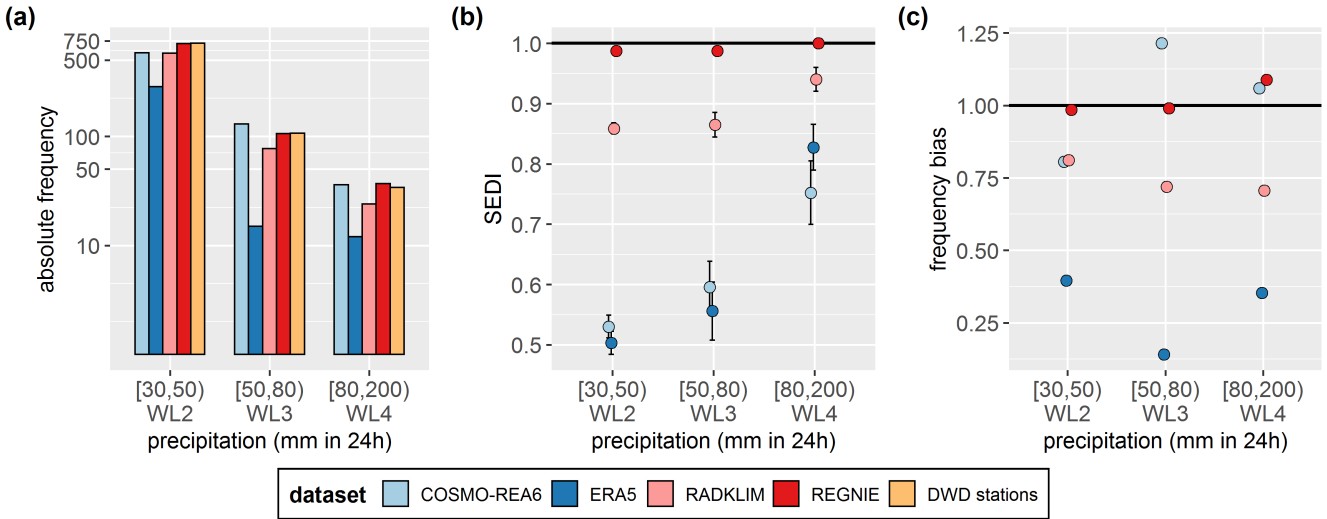

**Figure 2.** Evaluation of daily precipitation estimates from REGNIE, RADKLIM, ERA5 and COSMO-REA6 compared with DWD station observations (see Fig. 1.b). The selected period is 2001 to 2018, using the nearest neighbour grid cells to the DWD stations. (a) Histogram showing the absolute occurrences of precipitation within WLs 2, 3 and 4, (b) Symmetric Extremal Dependency Index (SEDI) and (c) frequency bias. Positive (negative) frequency bias indicates an overestimation (underestimation) of events, the black horizontal line represents the optimum value.

scale precipitation and relevant atmospheric variables (Martens et al., 2020; Hersbach et al., 2020; Yu et al., 2021) render this product suitable for the applications of this study. Here we use ERA5 to assess the large-scale conditions of moisture transport (Fig. 6).

## 2.3 Simulations

We use CPM simulations when observations are nonexistent or lowly resolved, as well as for sensitivity experiments.

### 2.3.1 Convection-permitting Weather Research and Forecasting (WRF) simulation for process-understanding

WRF model version 4.2.1 (Skamarock et al., 2021) at a convection permitting resolution of 1.5 km with 100 vertical levels is used. The model domain covers central Europe (Fig. 1a) and the simulation was initialized with the operational ECMWF analysis at 00 UTC on 29 June 2017 (12 hours before the event). Lateral boundary conditions were provided every 6 hours. The set up of the physical parametrizations is similar to Schwitalla et al. (2021) except that 1) the Mellor–Yamada–Nakanishi–Niino (MYNN) planetary boundary layer parametrization (Nakanishi and Niino, 2006, 2009; Olson et al., 2019) is applied, and 2) the microphysics scheme of Thompson and Eidhammer (2014) (TE2014) is used. Land cover maps have been updated using the high-resolution European Space Agency Climate Change Initiative (ESA CCI) data set (ESA, 2017). In addition, the green vegetation fraction and leaf area index (LAI) have been adjusted for 2017 by means of the Copernicus Global Land Service





portfolio 1 km resolution data sets (https://land.copernicus.eu/global/). Finally, high-resolution soil texture data from Poggio et al. (2021) (https://soilgrids.org) were used.

### 2.3.2 Convection-permitting COSMO-CLM simulations for conditional attribution

For the conditional climate-change attribution experiment (Sect. 3.6.1), high-resolution ensembles (17 members) of the event under present and pre-industrial conditions are simulated with the COSMO model in CLimate Mode (COSMO-CLM; Rockel et al., 2008). For the present climate, ERA5 reanalysis (Hersbach et al., 2020) is dynamically downscaled to 0.11° for 17 members over a pan-Europe domain (Fig. S4) using the domain-shift ensemble technique (e.g. Rezacova et al., 2009; Noyelle et al., 2019). All members are then further dynamically downscaled to a convection-permitting 0.025° resolution over a fixed

$461 \times 421 \times 50$ sub-domain. To create the pre-industrial ensemble, the warming signal since the pre-industrial period, here taken as 1850-1859 versus 2007-2016 (over the 0.11° domain), is first computed from a subset of 17 Coupled Model Intercomparison Project Phase 6 (CMIP6) models (O'Neill et al., 2016). The warming signal (Fig. 13a) is then subtracted from the ERA5 initial and boundary conditions of the 0.11° simulations (surface temperatures are modified based on the warming signal at the lowest model level). A similar procedure is repeated for soil temperature. The atmospheric moisture content is adjusted

based on the assumption that relative humidity remains constant. Pressure at the COSMO-CLM height-levels is adjusted by numerically integrating the hydrostatic balance equation downwards from the model top (Kröner, 2016). The 0.11° and 0.025° simulations are initialized on 28 June at 12:00 and 23:00 UTC, respectively, giving sufficient spin-up and adjustment time prior to the analysis period of 07:00 to 22:00 UTC the following day (chosen based on the analysis in Fig. S5). The attribution analysis consists of comparing the precipitation between the 0.025° ensembles with the pre-industrial ensemble as the refer-

ence state. COSMO-CLM model settings are as described in Meredith et al. (2021). Further details of the simulations and the CMIP6 models can be found in the supplementary material.

### 2.3.3 Sub-kilometre ICON simulations for aerosol sensitivity experiments

Simulations with the ICOsahedral Non-hydrostatic (ICON) atmospheric model were conducted to examine the possible role of anthropogenic aerosols in the analysed event. The model domain covered central Europe (Fig. 1), for which one-way nested

simulations at resolutions of 625 m and 1.25 km, respectively, are produced with 90 vertical levels. The model setup otherwise follows Costa-Surós et al. (2020). The simulation was initialized with the operational ECMWF analysis (IFS data) at 00 UTC on 29 June 2017 and lateral boundary conditions are provided every 6 h. For analysis, the daytime period between 06:00 and 20:00 UTC is chosen. In order to assess the role of cloud-active aerosols in the 2017 event, two simulations are carried out with two different imposed concentrations of cloud condensation nuclei, i.e. one with low concentrations, corresponding to

current conditions (CLN), and one with elevated aerosol concentrations (POL), corresponding approximately the peak aerosol concentrations over Central Europe observed in the mid-1980s.





### 2.4 Impacts and losses

To assess the overall losses caused by the 2017 event, we reviewed reports from the German Insurance Association (GDV), which published loss estimates for severe extreme precipitation events in Germany in the period 2002-2019 (GDV, 2018, 2020, 2021),

and other technical reports (Hydrotec et al., 2008). At the household level, we used losses, flood-duration and flood-depth data derived from household surveys in Berlin and Brandenburg (Dillenardt et al., 2021). We furthermore used survey data from private households in Hersbruck (affected in 2005), Lohmar (affected in 2005), Osnabrueck (affected in 2010) (Rözer et al., 2016) and Muenster (affected in 2014) (Spekkers et al., 2017). In the surveys, affected households were asked, inter alia, when they were last affected by heavy rain and what impact they suffered. The results of the surveys are based on the responses of

those households and are thus subjective.

### 2.5 Extreme value statistics

Extreme value theory is used to analyse or statistically model extreme values of a data set. A probability distribution function is fitted to a sample of extreme events. This allows a probability of exceedance $p$ for a given threshold $R$, e.g. a rainfall total, to be computed and expressed as a return period $t_{RP}$ using $t_{RP} = 1/p$. For example, a probability of exceedance p = 0.01 for

annual data can be interpreted as on average 1 event in 100 years.

In order to estimate these values reliably, a trade-off has been made in the past between the simplicity of a method, the return period size and the computational effort (Svensson and Jones, 2010). In hydrometeorology, block maxima using the GEV have been widely adopted to estimate return periods (Wilks, 2006; Svensson and Jones, 2010). Often the special case, the type I distribution, also known as the Gumbel distribution (Extreme Value Type 1), has been used (e.g. Grieser et al., 2007; Svensson

and Jones, 2010; Van den Besselaar et al., 2013; Maity, 2018; Piper et al., 2016). Characteristic of the Gumbel distribution is an exponential decay of the probability density function, meaning that only two free parameters of the GEV-fit have to be estimated. However, the Gumbel is not a heavy-tailed distribution and is characterised by constant skewness and kurtosis.

The block maxima series are obtained by splitting the sample into non-overlapping intervals of the same size and then taking the maximum value of each interval. Previous studies have demonstrated that the annual block maxima approach is suitable

for mid-latitude precipitation series (Kharin and Zwiers, 2000; Rust, 2009). The GEV distribution is given by Coles (2001):

$$G(R) = \exp\left(-\left\{1 + \xi\left(\frac{R-\mu}{\sigma}\right)\right\}^{-1/\xi}\right), \tag{1}$$

where $\mu$ denotes the location parameter, $\sigma$ the scale ($\sigma > 0$) and $\xi$ the shape parameter. In the limit $\xi \to 0$, we obtain the special case Gumbel distribution (Gumbel, 1958):

$$F(R) = \exp\left(-\exp\left(\frac{\mu-R}{\sigma}\right)\right). \tag{2}$$

This simplification allows estimation of the two free GEV parameters $\mu$ and $\sigma$ by means of the Method of Moments, which is less computationally intensive and has been proven useful in the past (e.g. Svensson and Jones, 2010; DWA, 2012; Piper et al., 2016). Return values $RV_{RP}$ (quantiles) and the associated return periods $t_{RP}$, or directly the probability of exceedances $p$, are





related to the GEV distribution as $t_{RP} = 1/(1 - G(RV_{RP}))$ (Coles, 2001). Thus, once $\mu$ and $\sigma$ are estimated from the data, the return periods can be directly derived from the extreme value models with Equations 1 and 2.

Additionally, we specify the uncertainty of the models by using 95 % confidence intervals obtained from 1000-fold boot-strapping (re-sampling with replacement) from the maxima series of the original data set (Efron and Tibshirani, 1993), follow-ing the ordinary non-parametric bootstrap percentile method of Mélèse et al. (2018). From these 1000 samples the 2.5 % and 97.5 % quantiles of the wanted property (either return periods or return values) indicate their confidence boundaries.

A more efficient usage of observations could improve the estimation of return periods. This can be achieved by including
precipitation maxima of different accumulation durations $d$. This way, information about short time scales (minutes to hours), for which observational records are typically short, can be derived, to a certain extent, from information of longer time scales (hours to days) by using the duration-dependency between time scales. Equation 1 is adapted to be duration-dependent ($d$-GEV) with 7 parameters, which covers the full range of durations considered. Duration-dependency is incorporated into the GEV in order to reduce assumptions about the underlying distribution and have a free shape parameter $\xi$ in the GEV (Coles,
2001). Another reason for using the GEV instead of one of their special cases, is that Gumbel, as well as the other two GEV special cases Fréchet and Weibull, require large data sets to fulfil the limiting theorem for large block sizes (Papalexiou and Koutsoyiannis, 2013). The d-GEV includes all three types and is a good choice if block size does not reach the asymptotic regime required for Gumbel. A duration-dependent GEV ($d$-GEV) as suggested by Koutsoyiannis et al. (1998) and refined by Fauer et al. (2021) is introduced by varying the characteristic parameters of the GEV:

$$G_d(R; d) = \exp\left( -\left\{ 1 + \xi \left( \frac{R - \mu(d)}{\sigma(d)} \right) \right\}^{-1/\xi} \right), \tag{3}$$

$$\sigma(d) = \sigma_0(d + \theta)^{-(\eta + \eta_2)} + \tau,$$

$$\mu(d) = \tilde{\mu}(\sigma_0(d + \theta)^{-\eta} + \tau),$$

$$\xi = \texttt{const.},$$

where $\tilde{\mu}$ is the re-scaled location parameter, $\sigma_0$ is the scale offset, $\theta$ is the duration offset, $\eta_1$ and $\eta_2$ are duration exponents and
$\tau$ is the intensity offset. Parameters were estimated with Maximum Likelihood Estimation (MLE), a flexible and efficient pa-rameter estimation method which is known to provide asymptotically unbiased and smallest-possible variant estimates (Coles, 2001; Davison and Huser, 2015). However, MLE is more computationally expensive than the method of moments. Jurado et al. (2020) justified the use of the MLE for duration-dependent extreme precipitation studies as the explicit consideration of the dependence between durations in a model leads to only marginal differences in estimation. The $d$-GEV has been recently
applied successfully by Ulrich et al. (2020).





# 3 Results

## 3.1 Meteorological situation

The 29 June 2017 occurred under the influence of an upper-level trough over western Europe (TrW pattern), present between 28 June and 01 July, with several short-wave surface lows developing on the northern side of the trough (Fig. 3). At 12 UTC (29 June), the core of the low-pressure was located between France and the British Isles showing relative topography values of 550 dam between 1000 hPa and 500 hPa (Fig. 3). Close to the ground, the interaction between two small-wave surface lows caused most of the precipitation collected over Berlin during 29 June. The first system, named Rasmund, remained quasi-stationary east of the British Isles between 28 June and 01 July, reaching values of the Pressure at the Mean Sea Level (PMSL) of 994 hPa (Fig. 3). This system originated from two converging surface lows coming from the Atlantic Ocean and France, respectively. The second surface low, Rasmund II, originated in the night of 29 June over central Europe (Czech Republic), displaced towards northern Poland over the course of 6 h, and showed PMSL of 990 hPa along the German-Polish border at 12 UTC (29 June). The cyclonic circulations of both surface lows caused convergence of cold air masses from the Atlantic with the warm and wet air masses from southern Europe (Gebauer et al., 2017).

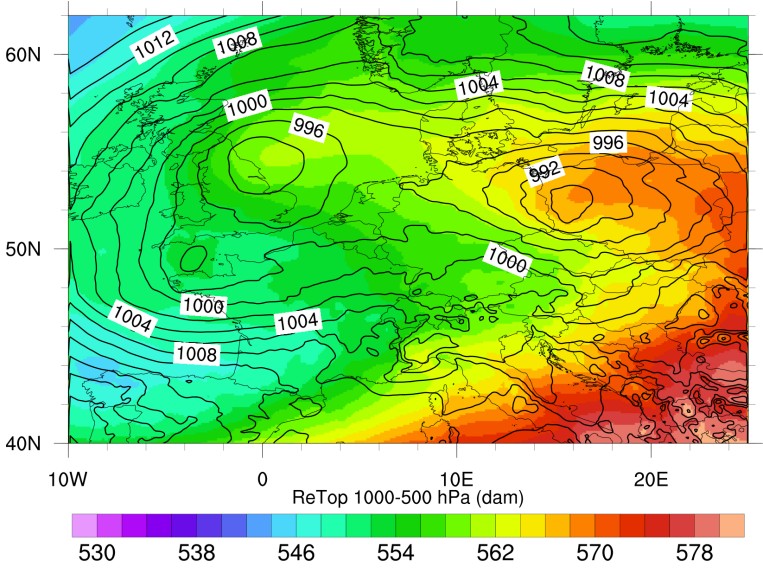

**Figure 3.** Mean Sea Level pressure (PMSL; contours) and relative topography between 1000 hPa and 500 hPa in dam (colour shading) on 29 June, 2017 12 UTC from the ECMWF analysis.

Figure 4 provides further insights into the evolution of the event based on the operational ECMWF analysis. The mesoscale circulation associated with the mid-tropospheric low Rasmund II (550 dam; Fig. 4a) shows a westerly flow around Germany turning into a southerly direction over Poland and eastern Germany with wind speeds of $10\,\mathrm{m\,s^{-1}}$ at 500 hPa (black arrows; Fig. 4a). Weak winds of less than $5\,\mathrm{m\,s^{-1}}$ also occur at 850 hPa (Fig. 4b). The Equivalent Potential Temperature ($\theta_e$) at 850 hPa





(Fig. 4b) shows values up to 305 K between the Mediterranean and central Europe, up to Poland and the Berlin metropolitan area. This is indicative of a high moisture availability and optimal conditions for associated stationary deep-moist convection.

During daytime (not shown), the strong low-level mid-day convergence together with integrated water vapour values of more than 40 mm led to extreme precipitation in the area. The strong low-level convergence was fostered by the counter circulations of Rasmund and Rasmund II. The low-level wind convergence, crucial for dynamic triggering, took place under the presence of moderate Mixed-Layer CAPE (ML-CAPE) of approx. 250 J kg$^{-1}$ and no Convective Inhibition (CIN). This was shown by the radio sounding deployed in Lindenberg, near Berlin, at 06 UTC 29 June 2017 (see Fig. S2).

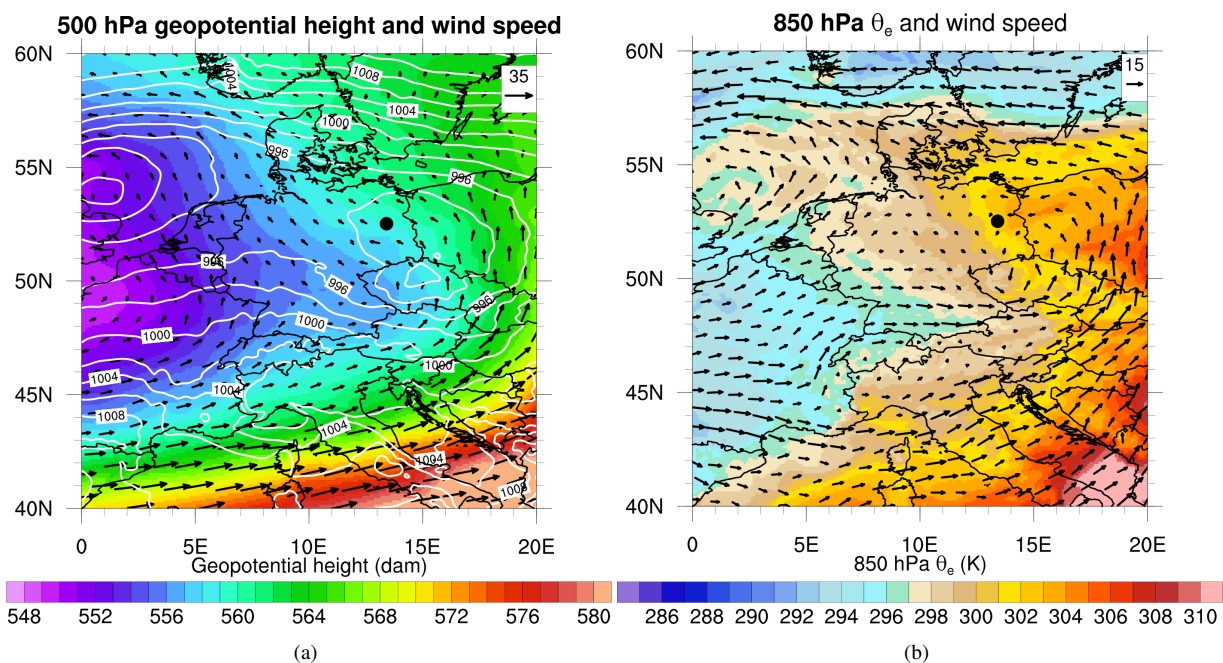

**Figure 4.** a) 500 hPa geopotential height (dam; colour) and wind speed (m s$^{-1}$) together with sea level pressure (hPa; contours) and b) 850 hPa equivalent potential temperature ($\theta_e$) and wind speed on 29 June, 2017 06 UTC from the ECMWF analysis. The black dot denotes the city of Berlin.

The diurnal development of low-level moisture was additionally investigated by means of a high-resolution WRF simulation (Sect. 2.3.1). Fig. 5 shows the time-height cross-section of water vapour mixing ratio (a) and rain water mixing ratio (b), for a grid point located northwest of the city of Berlin (black dot in Fig. 4), where the largest severe precipitation occurred. The WRF model simulates high water vapour mixing ratios of more than 13 g kg$^{-1}$ in the lowest 100 m above ground between 10 UTC and 19 UTC together with a strong low- to mid-level jet evolving after 13 UTC (Fig. 5a). This low-level jet is probably induced

by the temperature gradient between the colder and drier air masses in southwest Germany and the warm and moist air masses over northeastern Germany (Fig. 4a, Fig. S1). These factors led to a stronger pressure gradient and thus higher wind speeds. The precipitable water content over the period of interest varies between 40 mm and 44 mm, which is in accordance with the values



derived from the Lindenberg sounding (Fig. S2). The simulated rain water mixing ratios are very high, exceeding 3.5 g kg$^{-1}$ (Fig. 5b), potentially indicating a warm-rain type precipitation event (e.g. Song and Sohn, 2018) which is associated with a

strong downdraft of $\sim$ 5 m s$^{-1}$ (Fig. S3 of the supplementary material).

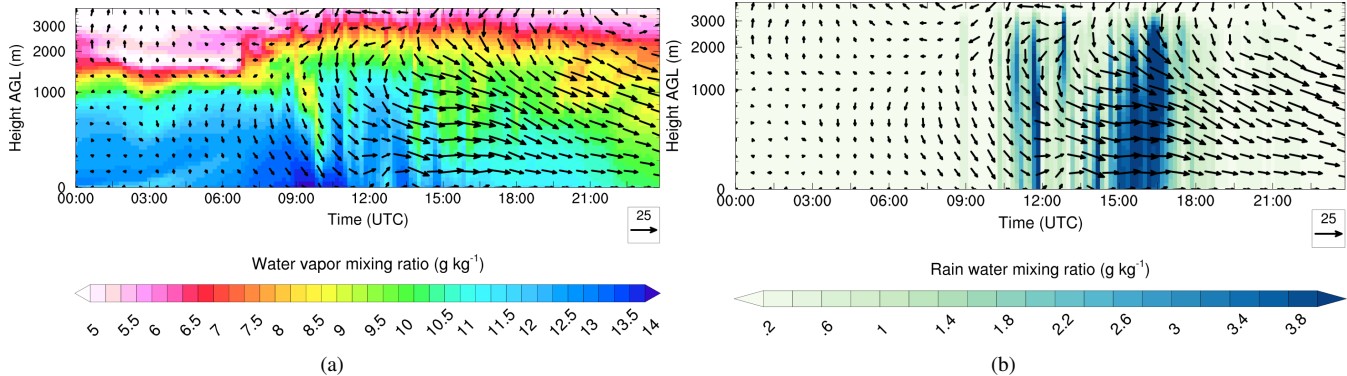

**Figure 5.** Time-height cross section of (a) water vapour mixing ratio (g kg$^{-1}$) and (b) rain water mixing ratio (g kg$^{-1}$) at grid point 13.2 °E and 52.85 °N (close to the maximum heavy precipitation location, near black dot in Fig. 4) simulated by the WRF model. The displayed period is 00 UTC until 23:45 UTC on 29 June 2017. Arrows denote the horizontal wind speed (m s$^{-1}$, see lower right of the plots for a length indicating 25 m s$^{-1}$) and direction at the different altitudes. Altitude is above ground level (AGL).

## 3.2   Lagrangian moisture source analysis

To analyse the origin of the atmospheric moisture that led to the large precipitation amounts during the event, we calculate Lagrangian backward trajectories following Sodemann et al. (2008). We calculate trajectories based on ERA5 reanalysis data (Hersbach et al., 2020), in a mid-European region around the center of maximum precipitation (red box in Fig. 6). We start

the trajectories from 1000 hPa to 200 hPa in steps of 50 hPa for every hour of 29 June 2017, with a horizontal grid spacing 80 km and going back 240 h in time. From all trajectories, we selected those that have a relative humidity of at least 80 % in the target box and for which the specific humidity decreases during the last time step (precipitation) (Sodemann et al., 2008; Grams et al., 2014). Along each selected trajectory (4,080 out of 15,074 in total), moisture uptake has been computed based on hourly specific humidity increases in the planetary boundary layer associated with evapotranspiration from the surface. The

boundary layer height is available as a diagnostic model variable in the ERA5 data set and, again following Sodemann et al. (2008), is multiplied by a factor of 1.5 to account for potential uncertainties in this diagnostic estimate. These uptakes have been weighted according to their contribution to the precipitation (moisture loss) at the target location, taking into account that earlier moisture uptake may contribute less due to precipitation loss on their way to the target region. With this approach, about 49.8 % of the precipitating moisture for the day of 29 June can be traced back to its moisture source.

For this event, land masses are the main sources of moisture uptake with a total contribution of 82.9 %, whereas only 17.1 % of the precipitating moisture evaporated over the ocean (of the identified 49.8 % precipitating moisture; Fig. 6). The eastern
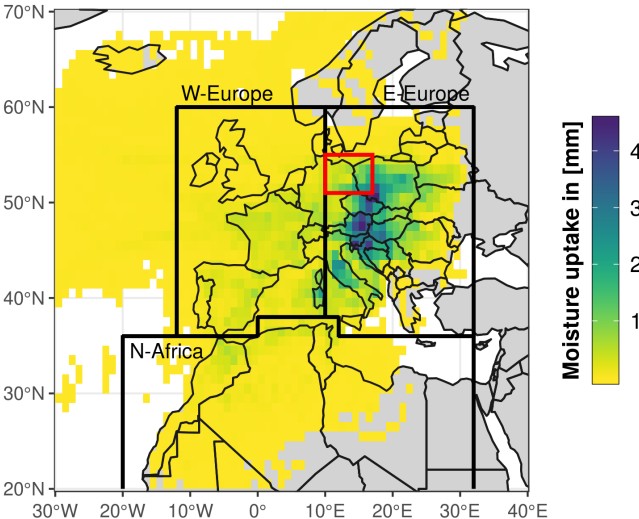

**Figure 6.** Moisture uptake within the planetary boundary layer, calculated by Lagrangian backward trajectories using ERA5 reanalysis data. Hourly initialised trajectories on 29 June 2017 (0 UTC to 23 UTC) at the region of heavy precipitation (red box) from 1000 hPa to 200 hPa in steps of 50 hPa and 80 km horizontal distances, going back 240 h in time, are used to compute moisture uptake based on humidity changes, if they are associated with precipitation in the target region.

European region (around $10°$E to $32°$E and $37°$N to $60°$N) is by far the main source of moisture uptake (63.0 %). Additionally, the moisture uptake in this region is relatively evenly distributed, ranging from Poland (east of the precipitation event) towards Croatia and Italy, with a maximum moisture uptake of about 4.6 mm per single grid point. Other, but less important, land

moisture sources are the western European region (around $12°$W to $10°$E and $37°$N to $60°$N), with a contribution of 13.9 %, and the northern African region (around $20°$W to $32°$E and $20°$N to $37°$N), with a contribution of 5.9 %. The oceanic moisture sources are primarily the Mediterranean Sea (11.9 %) and Atlantic Ocean (4.6 %), but these play a minor role compared to the moisture uptake over land. A similarly important role of moisture recycling from land sources has been found previously for an extreme precipitation event in eastern Europe in May 2010 (Winschall et al., 2014) and for the central European floods in June

2013 (Grams et al., 2014; Kelemen et al., 2016). In the case of the 2017 event studied here, moisture recycling likely happened on relatively short time scales of 1-2 days, as June 2017 was generally dry, but northeastern Italy, Slovenia, Austria and southeastern Poland were affected by convective precipitation on 28 June. The moistening of the soil due to prior precipitation is thus hypothesised to be an important precondition for the Berlin event.

### 3.3 Observed lightning activity and accumulated precipitation

To better understand the temporal evolution of the convective development, the spatial distribution of lightning strikes in northeastern Germany is presented (Fig. 7a). The combination of low-level wind convergence associated with the surface lows and the warm and moist air masses transported into the region (Fig. 4) favoured thunderstorm activity in the border


region between northeastern Germany and Poland. Already in the morning hours starting at 5 UTC, an initial convective line moving westward from Poland towards the city of Berlin produced first cloud-to-ground lightning along the border (Fig. 7a).

The system strengthened and hit Berlin mainly between 9 and 10 UTC. Afterwards, the system weakened and moved only slightly further west due to weak upper-level flow. It remained relatively stationary over the greater Berlin area and west of it until noon (Fig. 7a). Upstream of the system, a second thunderstorm line followed, crossing the border between Poland and Germany between 14 and 15 UTC. In the late afternoon, the direction of the convective cells changed and they were transported northward with the weak upper-level flow. In the evening of 29 June 2017, Mecklenburg-Western Pomerania was particularly

affected, as well as the Baltic Sea.

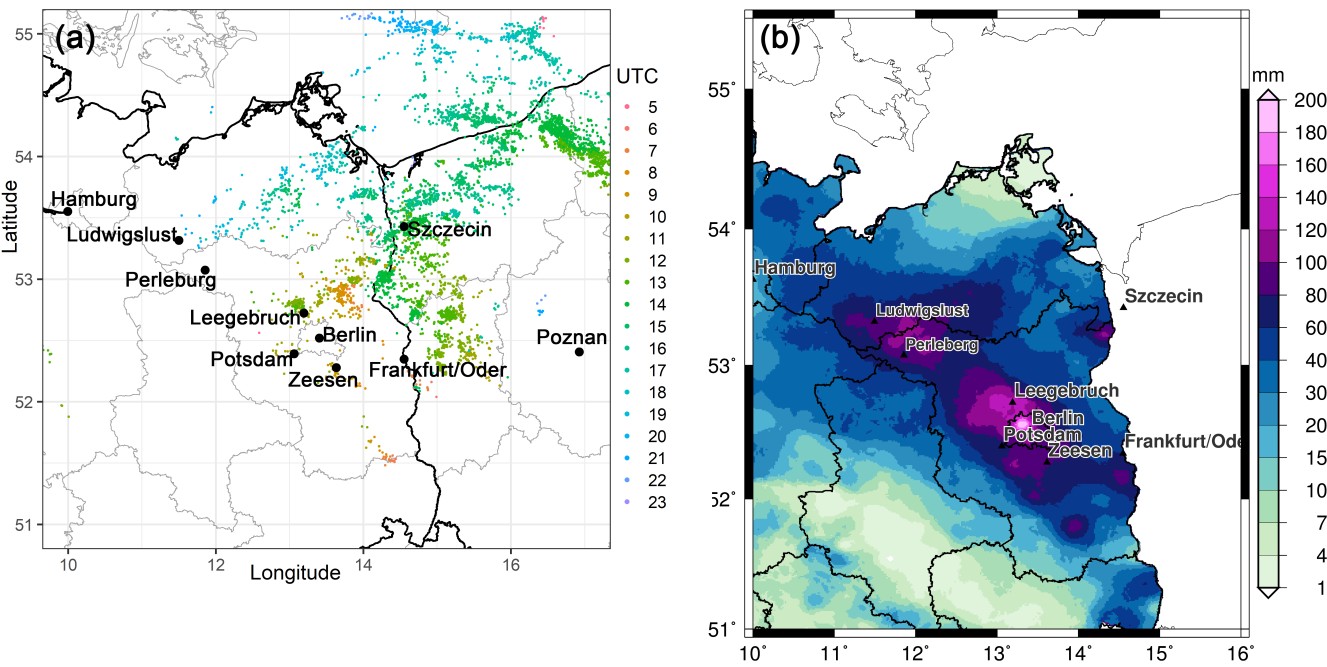

**Figure 7.** (a) Cloud-to-ground strokes from the EUCLID data set, colour-coded according to the time of occurrence from 5 UTC on 29 June 2017 to 0 UTC on the following day and (b) 24-hour precipitation totals (mm) during the study period (29 June 2017 06 UTC to 30 June 2017 06 UTC) from REGNIE .

Due to a weak upper-level flow (Fig. 4a), the thunderstorms were associated with low propagation speeds leading to high local rain rates. Figure 7b shows daily precipitation totals with values up to 200 mm. The convectively enhanced precipitation totals fell mainly in the course of 12 hours in the German states of Brandenburg (BB), Berlin (BE), and southern Mecklenburg-Western Pommerania (MV). Very high values above 100 mm were recorded in and northwest of the city of Berlin as well as

in an area between Ludwigslust (MV) and Perleberg (BB). For example, Berlin-Tegel (BE) recorded 24-hour precipitation of 196.9 mm, while Zeesen (BB) registered 149.9 mm (Fig. 7b and Table S1 in the supplementary material). In many places, more




precipitation was measured within 24 hours than the climatological mean for the whole month of June (Wandel, 2017). In the following, we assess the impact that this extreme atmospheric situation had on infrastructure and private households.

### 3.4 Impacts during the 29 June 2017 case

The extreme precipitation event on 29 and 30 June 2017 heavily impacted the metropolitan area Berlin-Brandenburg ($\sim$4.5 Mio. inhabitants). In total, the event caused €60 Mio. in insured property loss (in 2019 prices), most of which occurred in Berlin and Brandenburg (GDV, 2018). This makes it the most damaging extreme precipitation event in Berlin and the Oberhavel district, Brandenburg, in the period 2002-2017, for which insurance data provide coverage. Insurance data of the GDV show that 1.8 % of the buildings in Berlin incurred damages during the event, with an average loss of €6,830. The Oberhavel district was

more heavily affected, with 5.3 % of the buildings damaged and an average loss of €10,550 (GDV, 2021). In Berlin, the long-lasting rainfall overloaded the sewer system, resulting in widespread inundation that caused disruption of traffic (i.e., blocked roads), flooded basements and underground stations, as well as intangible consequences including restrictions in daily routine and mental discomfort (GDV, 2018; Berghäuser et al., 2021). The high number of rainfall-related missions on these two days forced the Berlin fire brigade to declare an 'exceptional weather situation status' (Kox and Lüder, 2021). The small municipality

of Leegebruch ($\sim$6,800 inhabitants, located 40 km north of Berlin in the Oberhavel district) was even more severely affected due to its location in a topographical depression and a typically high groundwater table. The resulting inundation cut off the settlement from its surroundings, affected 40 % of the municipality, and persisted for several weeks (GDV, 2018).

**Table 1.** Pluvial flooding events where surveys were conducted at affected households. Five events were surveyed at seven locations. The start and end dates, the affected area, the maximum daily precipitation, and the affected population are obtained from the CatRaRe database (Sec. 2), based on RADKLIM precipitation data. The overall losses within the municipality (*) are from GDV (2020) and refer to insured losses, while loss values with a (**) are from Hydrotec et al. (2008) and refer to total losses (non-insured and insured losses). Losses are referred to constant 2019 prices. Note that Hersbruck was hit by two events in close succession, which are listed separately.

| City | Date Start yyyy-mm-dd | Date End yyyy-mm-dd | Aff. Area km$^2$ | Max. Prec. mm | Population 10$^3$ people | Losses Mio. € | Surveys |
|---|---|---|---|---|---|---|---|
| Berlin | 2017-06-29 10:50 | 2017-06-30 10:50 | 31,661.4 | 161.9 | 6,529 | 60 (*) | 28 |
| Leegebruch | 2017-06-29 10:50 | 2017-06-30 10:50 | | | | | 88 |
| Muenster | 2014-07-28 13:50 | 2014-07-28 22:50 | 1,117.3 | 175.8 | 594 | 330 (*) | 447 |
| Greven | 2014-07-28 13:50 | 2014-07-28 22:50 | | | | | 63 |
| Osnabrueck | 2010-08-26 04:50 | 2010-08-27 04:50 | 13,426.9 | 163.9 | 4,975 | 90 (*) | 100 |
| Hersbruck | 2005-06-29 05:50 | 2005-06-29 07:50 | 53.6 | 42.8 | 20 | 4 (**) | 111 |
| Hersbruck | 2005-06-29 20:50 | 2005-06-29 22:50 | | 39.8 | 19 | | |
| Lohmar | 2005-06-29 00:50 | 2005-06-30 00:50 | 2,669.7 | 100.9 | 2,571 | 3.5 (**) | 62 |

Table 1 compares the meteorological severity, the impacts and the number of surveyed households between the Berlin-Leegebruch-2017 event and the events in Muenster and Greven (impacted in 2014), Osnabrueck (impacted in 2010) and





Hersbruck and Lohmar (both impacted in 2005). The meteorological indicators (derived from CatRaRe) show that the Berlin-Leegebruch-2017 event was characterised by its large spatial extent (31,661 km$^2$) and long rainfall duration (approx. 24 hours). As shown in Sect. 3.1, the long rainfall duration was caused by the slow propagation of the convective system given the weak mid-tropospheric winds. The maximum accumulated precipitation in Berlin, shown in Table 1, somewhat differs from the REGNIE observations (Fig. 7b) since the values of CatRaRe are based on the RADKLIM RW 1-hour product that uses a different

gridding method and data source. The other events affected considerably smaller areas and, in the case of Muenster-Greven-2014 and Hersbruck-2005, persisted for a shorter period. The maximum precipitation depth in Berlin-Leegebruch-2017 was exceeded by the events in Muenster-Greven-2014 and Osnabrueck-2010. The aggregated event loss in Berlin-Leegebruch-2017 was lower than the losses caused by the extreme rainfalls in Muenster-Greven-2014 and Osnabrueck-2010, although the Berlin-Leegebruch-2017 event affected the largest number of people. The events in Hersbruck-2005 and Lohmar-2005 exhibited less

intense rainfall and substantially smaller losses.

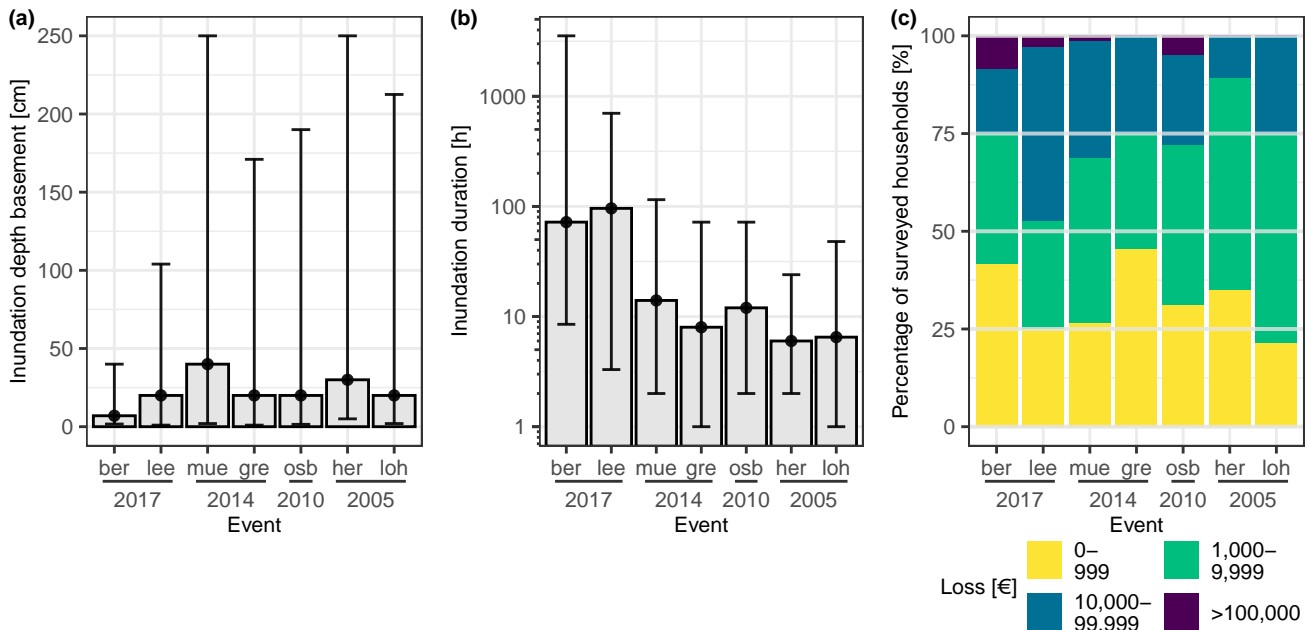

**Figure 8.** Comparison of (a) inundation depth, (b) inundation duration and (c) building loss between pluvial flood events in seven municipalities as reported in the surveys Berlin-2017 (ber), Leegebruch-2017 (lee), Muenster-2014 (mue), Greven-2014 (gre), Osnabrueck-2010 (osb), Hersbruck-2005 (her), Lohmar-2005 (loh). Points indicate median values, while whiskers show 5 % and 95 % percentiles.

Figure 8 allows for a more detailed view of the impacts and related mechanisms by looking at the surveyed flood indicators and monetary losses at a household level. The figure panels show the distributions of the inundation depth in the basement, the inundation duration and the building losses that were reported by private households affected by the respective pluvial flood events. The highest basement inundation depths occurred in Muenster-2014 and Hersbruck-2005. In Leegebruch-2017 and

Berlin-2017, basements were inundated less severely than in the other events. The inundation durations in Berlin-2017 (median



of 72 hours) and Leegebruch-2017 (median: 92 hours) exceed the inundation durations of the other events considerably, which do not exceed 24 hours (Fig. 8b; note the log-scale). Monetary losses to buildings were particularly large in Leegebruch-2017, Muenster-2014 and Osnabrueck-2010, where approximately 75 % of the surveyed households reported a loss of €1000 and more. In addition, building loss exceeded €9,999 in more than 25 % of the cases and the largest loss category is relatively

abundant. In Leegebruch-2017 almost half of the surveyed households suffered building loss in the largest two categories (≥€10,000).

In summary, the Berlin-Brandenburg-2017 was very extreme with respect to precipitation and flood duration, large spatial extent and extraordinary monetary losses. The joint evaluation of the CatRaRE and survey data reveals no uniform relationship between precipitation indices (intensity, duration, extent) and resulting losses across the group of study cases. In contrast, flood

characteristics at the affected buildings (inundation depth and duration) can explain much (although not all) of the incurred losses. Ultimately, flooding in urban areas is a complex process that depends not only on the meteorological nature of an event but also on the local characteristics of the terrain defined by topography, land use, sewer system capacity and operability and hydro-geology, as well as on socioeconomic conditions such as settlement structure, building codes and private risk-mitigation.

### 3.5 Probability of exceedance and severity

In this section we quantify the return periods and severity for this event.

#### 3.5.1 Probability of exceedance (return periods)

We estimate return periods using the Gumbel distribution (Eq. 2), the GEV distribution (Eq. 1) applied to spatially averaged data and a duration-dependent GEV (Eq. 3; see Sect. 2.5). While each approach has benefits and shortcomings (e.g. available data set, computing time), they all show that the event analysed was a rare event.

The Gumbel distribution (Eq. 2) is used to estimate exceedance probabilities and associated return periods per grid-point $(1 \times 1\,\mathrm{km}^2)$ for 24-hour precipitation totals in Germany based on a 70 year time series (REGNIE; Fig. 9). The results show return periods of more than 200 years between Ludwigslust (MV), Perleberg (BB) and Berlin; covering a total area of $8.729\,\mathrm{km}^2$. Past work has shown how estimated return periods that are much longer than the observational record exhibit large uncertainty (Makkonen, 2006; Grieser et al., 2007). This is why we truncate all return periods above 200 years (Fig. 9), acknowledging that

the 70 year observational record might be too short to provide accurate values for such long return periods.

The second approach, consisting of fitting a GEV distribution to spatial averages of different size also showed very large return periods, however decreasing with increasing size of the considered areas. The rationale behind this analysis is that return periods at individual grid points may not best-characterise the event and its associated impacts since an event with a very long return period at a given site can have a small spatial extent. Such a situation will lead to less flooding than an event with

similar precipitation intensities which is spatially larger. Such spatial averaging of observations is also required for validation of simulated return periods in climate models and assessment of the model biases (see e.g. Philip et al., 2020).

For this approach we fit a GEV distribution (Eq. 1) and estimate return periods of the yearly block maxima of REGNIE precipitation data from 1951 to 2020, spatially averaged over three boxes of different size (brown, grey, and pink boxes in Fig. 9





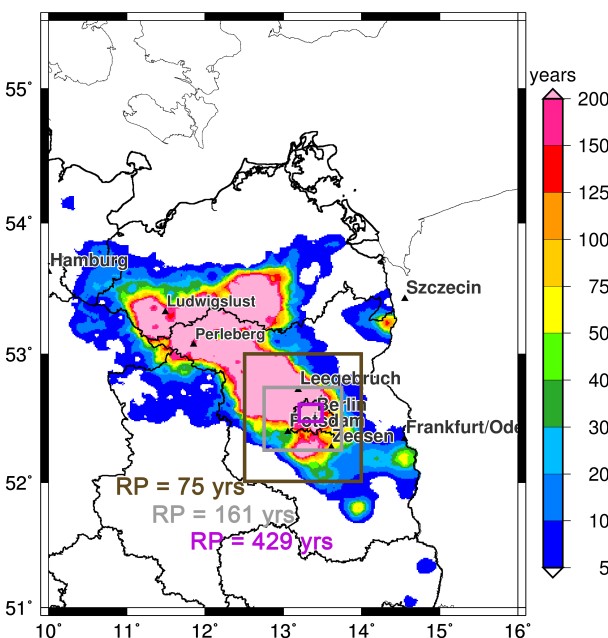

**Figure 9.** Return periods of the 24-hour precipitation totals based on REGNIE using a Gumbel distribution (Extreme Value Type 1; reference period: 1951–2020). Note: All return periods above 200 years are shown uniformly in one colour due to statistical uncertainty (see text for more information). Coloured polygons (in pink, grey, brown) including the associated averaged return periods (RP) indicate three different-sized averaging areas explained in Table S2 and in the text.

and Table S2 in the supplementary material). The results show that the return period of the event decreases with increasing

box size. At the finest analysed scale ($340\,km^2$; pink polygon), the extreme event has a return period of over 420 years, which decreases to 75 years for the largest area ($11,100\,km^2$; brown polygon) due to the regional scale of this extreme event. Compared to other historical events, the precipitation of June 2017 was the most extreme observed event in the time series when considering the smallest analysed region, however, an event with a larger spatially-averaged precipitation sum and thus a larger return period was detected on 8 August 1978 in the two larger areas (grey and brown boxes). In the supplementary

material, the return period of the 2017 event is compared to the July 2021 heavy rainfall event impacting Western Europe.

One useful approach to reduce uncertainty in the estimation of return periods from temporally short observational records is using different accumulation durations between 1 hour and 3 days (Eq. 3), however at the expense of more demanding computing power. The $d$-GEV model (Eq. 3) is applied to grid point intensity from RADKLIM RW ($2001-2020$; Fig. 10a). For this dataset, the estimated $d$-GEV shape parameter $\xi$ has a median value of 0.24. Accumulations (duration $d$) of one day showed

very high return periods ($>800$ years) for seven grid points in north-western Berlin. Over this area, most grid points showed return periods between 50 and 200 years. The small grid box size of $1\,km \times 1\,km$ could explain the very high return periods in cases of statistical outliers. On shorter time scales of 8 hours and 1 hour, north-western Berlin shows lower return periods, between 10 and 100 years for the two former temporal aggregations and between 2 and 10 for the latter (Fig. 10.a). The short


time range of historical RADKLIM data (20 years), however, limits the reliability of these spatially resolved return period
estimates.

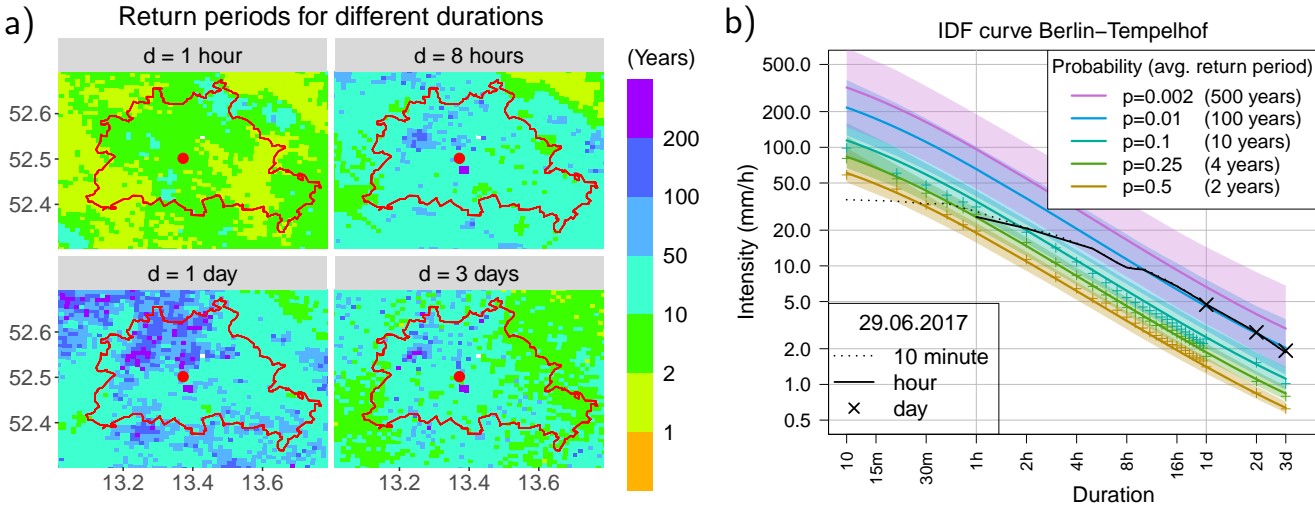

**Figure 10. a):** Return periods of this event's intensity from gridded RADKLIM ($1 \, \text{km} \times 1 \, \text{km}$) data using a duration-dependent GEV distribution. Red dot: position of DWD weather station Tempelhof, used to construct the IDF curve in (b). **b):** IDF curves from station-based data. The coloured lines show return periods from the $d$-GEV model, the black line indicates intensity of the event at 29 June 2017 for different temporal measurement resolutions. Empirical quantiles are denoted as coloured "+". 95 % confidence intervals come from 1000-fold bootstrapping.

To overcome the short time span of the RADKLIM dataset, we implement a duration-dependent GEV model (Eq. 3) on DWD ground station data at Berlin-Tempelhof (Fig. 1). The return levels (quantiles) for any exceedance probability and time scale (duration $d$) are presented in Intensity-Duration-Frequency (IDF) curves (Fig. 10). This station was chosen because data records are longer than the RADKLIM data. 10-minute aggregations span 1995 to 2020 and daily aggregations cover the period
1948 to 2020. The longer time series lead to shorter and—due to the reduced uncertainty—potentially more plausible return periods. The analysis shows return periods of 100 years for durations above than 10 hours, 114 years for daily aggregations. However, confidence intervals in the IDF curves also remain large (Fig. 10b).

All used approaches highlight that the analysed event was rare, with return periods longer than 100 years in the Berlin area. The efficient use of information from the observation data (accumulation periods from 10 minutes to 3 days) lead to more
plausible return period estimation from the $d$-GEV model, compared to Gumbel using daily accumulations. However, long data records with high measurement frequency and enough computing resources are not always available, in which case a non-duration-dependent model using daily data could be a better choice. Finally, including a climate change signal into the $d$-GEV estimation (Ganguli and Coulibaly, 2017) could improve the results, since the assumption of a non-changing climate leads to over-estimated return periods for events that become more likely with climate change.





### 3.5.2 Severity

In addition to the estimation of return periods, we assess how extreme the 29 June 2017 event was in terms of severity. We use the Precipitation Severity Index (PSI; Caldas-Alvarez et al., 2022) to detect extreme precipitation events according to three different but complementary characteristics of heavy precipitation: intensity, spatial extent and persistence. The PSI in its current form is an adaptation of the Storm Severity Index (SSI; Leckebusch et al., 2008; Pinto et al., 2012) and is a unitless index that indicates the degree of daily precipitation severity with respect to a predetermined climatological threshold (in our case the $80^{th}$ percentile). Large PSI values represent high intensity, geographically extensive and temporally persistent precipitation events.

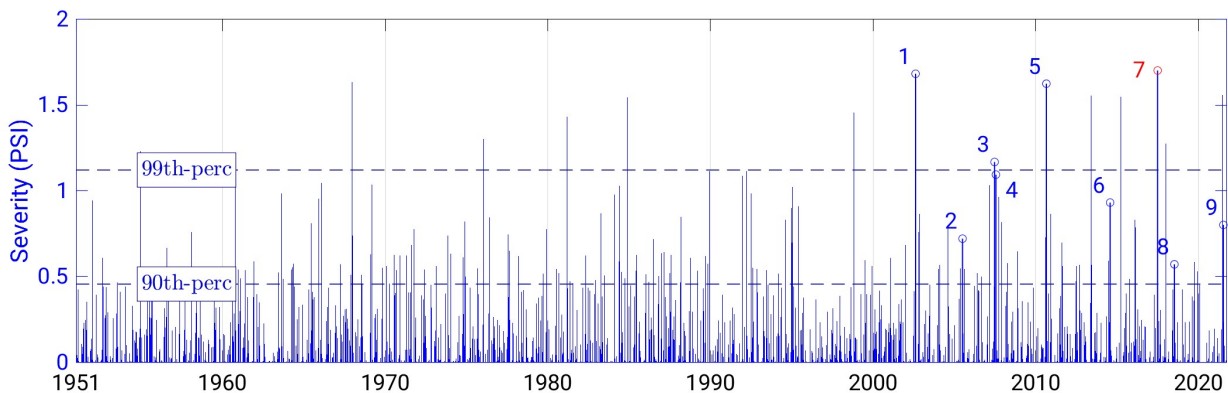

**Figure 11.** Temporal evolution of the PSI (blue bars). The results are based on REGNIE daily precipitation observations between January 1951 and September 2021. The 99-percentile and 90-percentile of daily PSI values are represented by the dashed blue horizontal lines. Numbered circles highlight nine events analysed in Table 1 or in Fig. 12 as well as other historical events. These are (1) the Saxony floodings on the 12 August 2002; (2) the 29 June 2005 event, affecting Hersbruch and Lohmar; (3) the 21 June 2007 and the (4) 22 July 2007 events affecting Germany in its totality; (5) the 26 August 2010 event in Osnabrueck; (6) the 29 July 2014 event in Muenster and Greven; (7) the 29 June 2017 event (red); (8) the 12 July 2018 causing flooding in Berlin; and (9) the Ahr flooding on 14 July 2021.

Figure 11 shows the temporal evolution of daily PSI values between January 1951 and September 2021 from daily REGNIE observations. The PSI values of 9 historical events from Tab. 1 and Fig. 12 are shown. The PSI analysis shows that, in terms of meteorological severity, the 29 June 2017 event around Berlin was the $29^{th}$ most severe event in the 1951-2021 period, well above the $99^{th}$-percentile of the climatology (Fig11). The severity of the 29 June 2017 case was found to be 2.3 and 1.8 times larger than the 29 June 2005 event in Hersbruch and Lohmar (Tab. 1; number 2 in Fig. 11) and the 29 July 2014 event in Muenster and Greven (Tab. 1; number 6 in Tab. 1), respectively. The 26 August 2010 event in Osnabrueck (Tab. 1; number 5 in Fig. 11) is the only event of those assessed in the household surveys with a similar meteorological severity and a similar extent of the caused damages (€90 Mio.; Tab. 1). Compared to other events, the 2002 event in Saxony causing large flooding (number 1 in Fig. 11) showed the most similar meteorological severity. The Ahr flooding in July-2021 (number 7 in Fig. 11)




had a PSI value 2.1 lower than the 29 June 2017 event due to the weaker grid point intensity (131 mm compared to 196.9 mm) and, especially, to the lack observations over affected areas in Belgium and the Netherlands in REGNIE.

To further classify the severity of the 29 June 2017 in the climatology we implement a simple cell tracking algorithm, originally developed for isolated convection (Steinacker et al., 2000; Purr et al., 2021) on 5 min, 1 km radar (YW RADKLIM), for the period 2001 to 2020 over Germany.

Figure 12 shows a scatter plot of events in that period where the number of detected tracks is compared to their mean length. Events are divided into two groups, depending on whether a prevailing wind direction and cyclonic circulation was present (red) or not (blue). The only event with more cells triggered than 29 June 2017 is the 12 August 2002 event, which caused the

historical floods in Saxony (Kreibich et al., 2007). Also outstanding with regard to the number of convective cells triggered were the events on 21 June 2007, 22 July 2007 and 12 July 2018, where flooding was reported for northern Bavaria, west and south-west Germany and Berlin, respectively (e.g., Kaiser et al., 2021). The former two events showed a strong meteorological severity, with a PSI in the 99th-percentile of the climatology and the latter showed a moderate severity with a PSI close to 0.6. Figure 12 also shows the inverse relationship between the length of the tracks and their number, i.e., the larger the number of

cells triggered, the shorter their mean length. Moreover, Fig. 12 shows that a large number of extreme events in the 2001 to 2020 period belong to the XXCCW weather type (Bissolli and Dittmann, 2001).

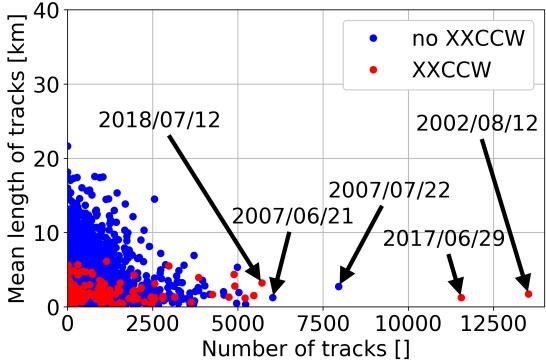

**Figure 12.** Cell track characteristics (mean length and number of tracks) for the research area Berlin and surroundings for days from 2001 to 2020. Colour coding identifies if days have been classified as weather type 'no prevailing wind direction, cyclonic circulation in 950 and 500 hPa and above-average humidity content of the troposphere - XXCCW' or another type (not XXCCW). Arrows indicate days with flooding events in Germany.

## 3.6 Extreme event attribution

A full understanding of an observed extreme event requires addressing the question of how the event relates to anthropogenic influences. This is particularly important with respect to climate-change communication with stakeholders and the general

public. The field of extreme event attribution seeks to address such questions by examining whether, and to what extent, anthropogenic influences may have affected the severity and/or frequency of a specific extreme.

### 3.6.1 Conditional event attribution

For conditional attribution, the modelling approach involves simulating an event under present-climate conditions and then repeating the simulation with modified boundary conditions. This modification consists of subtracting the vertical thermodynamical climate-change signal (Lackmann, 2015; Pall et al., 2017) and is, thus, particularly amenable to high-resolution (convection-permitting) regional-model experiments, which have many advantages for modelling extreme precipitation (Meredith et al., 2020, 2021; Stevens et al., 2020). The modelling approach is, in essence, an adaptation of the surrogate global-warming method (Schär et al., 1996; Hibino et al., 2018; Kröner et al., 2017). The results presented here are based on the model simulations described in Section 2.3.2. Analysis is performed over an area of $81{,}520$ km$^2$ centred on the event location. Model evaluation and technical details are found in the Supplementary Material.

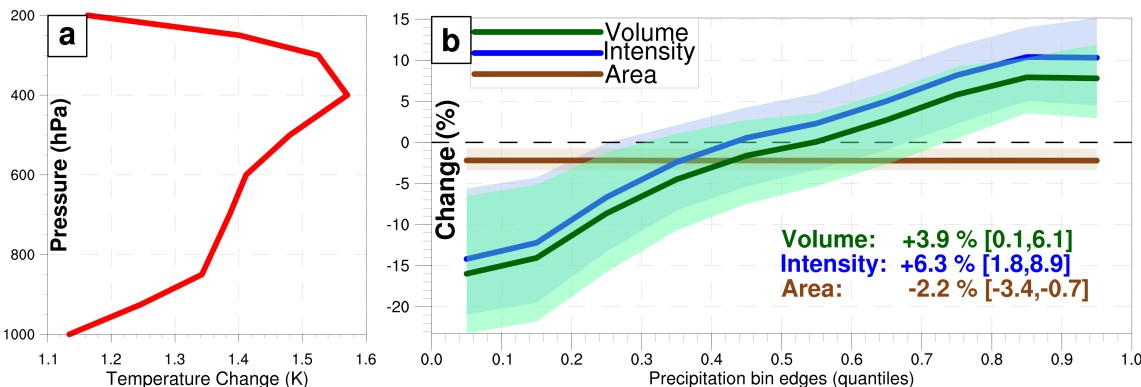

**Figure 13.** (a) Vertical warming signal and (b) response of event precipitation to warming signal. The warming is based on a subset of 17 CMIP6 models. The x-axis in (b) shows precipitation bins which are delineated by quantiles of the precipitation intensity distribution; the changes in area and volume represent the changes associated with precipitation in these bins. Based on the Wilcoxan-Mann-Whitney test, the total changes given within (b) are all statistically significant with at least $p < 0.02$. Shading denotes the 95 % confidence intervals based on 1,000 bootstrap resamples.

The thermodynamical climate-change signal is vertically heterogeneous, with stronger warming in the mid and upper troposphere (Fig. 13a). Based on a mean tropospheric ($1{,}000 - 300$ hPa) temperature of $\sim 269$ K and warming of 1.36 K, this implies an increase in saturation vapour pressure of $\sim 10.5$ % (Bolton, 1980), which exceeds the increase in precipitable water we find (7.5 %, time average; not shown).

In our present-climate simulations, we find a low, though statistically significant ($p < 0.02$), increase in total precipitation volume of 4 %. This can be better understood by analysing both changes in the precipitation distribution and physical characteristics of the convective system (Fig. 13b). Changes in total precipitation volume associated with a given quantile of precipitation intensity increase as the quantiles become higher. Indeed, at the lowest intensity quantiles, the associated precipitation volumes are found to decrease, which partly offsets the volume increases associated with the higher intensity quantiles (Fig. 13b). By breaking the total precipitation volume into its area and depth components, it can be seen that the change in precipitation in-
tensity shows a signal very similar to that of the total precipitation volume. The spatial extent of the precipitation, meanwhile, decreases (-2 %) in the warmer climate, in line with the results of Wasko et al. (2016) and Armon et al. (2022). It is thus clear that it is not changes in the spatial extent of the system, but rather higher local intensities which drive the increase in total precipitation volume. The intensities increase for, approximately, the upper half of the precipitation distribution, peaking at a

10.4 % increase for the highest quantiles. This increase exceeds the increase in precipitable water (see above), implying that local moisture convergence was an important factor for the most extreme intensities.

It is worth noting that, based on the average tropospheric warming signal (see above), the intensification of the highest quantiles corresponds (almost exactly) to what would be expected from the Clausius-Clapeyron relation. Had the precipitation-temperature scaling been computed using the lower troposphere or the 2-m temperature instead, a super Clausius-Clapeyron

increase would have been found. Currently, it is still a matter of debate as to which temperature is most appropriate to use when computing the scaling rate (Drobinski et al., 2016; Formayer and Fritz, 2017). It is plausible that the total precipitation would not have shown an attributable change but that the most intense quantiles would have. This insight is also of relevance for impact studies: the effect of climate change was found to be dependent on the spatial scale of interest. We conclude by stating that climate change since the pre-industrial era served to increase the magnitude and, in particular, the highest precipitation

intensities of the event.

### 3.6.2   Role of Aerosols

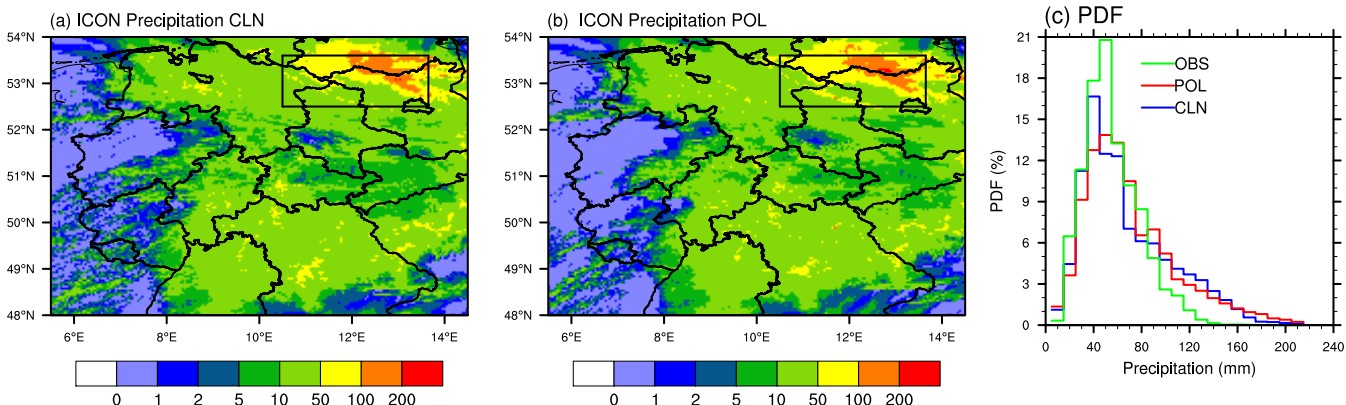

**Figure 14.** Results from the 635 m ICON simulations. Distribution of precipitation ($mm\,d^{-1}$) from (a) the factual CLN simulation and (b) the counterfactual POL simulation. (c) PDF of precipitation intensities in the region of interest denoted by the rectangle in a) and b), for the factual (CLN) and polluted (POL) simulations and the observations (OBS) from the RADKLIM precipitation observations.

The role of cloud-active aerosol within the 2017 event was examined using simulations with the ICON model shown in Fig. 14 (see Sect. 2.3.3). The simulated distribution of precipitation is compared with RADKLIM for the period from 06 UTC 29 June 2017 to 06 UTC 30 June 2017 (Fig. 14). In order to assess the importance of aerosols for this extreme event, two

simulations are performed with different aerosol number concentrations. The first one has factual concentrations representative



of the 2017 event, which is a comparatively clean situation (CLN). The second one, has aerosol concentrations representative of the peak-aerosol conditions in the mid-1980s, or polluted scenario (POL).

The results show that POL simulates an increase in cloud number and cloud mass due to the increased aerosol loading (not shown). This is particularly seen in the heavy precipitation intensities exceeding $150\,\mathrm{mm\,d^{-1}}$ (Fig. 14c). These would have been

higher by 70 % in the high-aerosol conditions as in the 1980s (4.3 % of grid points exceed $150\,\mathrm{mm\,24\,d^{-1}}$ in the high-aerosol case compared to 2.3 % in the low-aerosol simulation).

## 4 Conclusions and outlook

HPEs with high impact are gaining attention in the scientific community and the general public. Events such as the one assessed here, affecting the Berlin metropolitan area in June 2017, or the Ahr event in western Germany in July 2021 (Kreienkamp

et al., 2021) are evidence of the tremendous damaging potential of this atmospheric phenomenon. Multidisciplinary studies, combining expertise from different research fields are a powerful means to provide novel information about interconnected aspects of extreme precipitation. Here we used the synergy of the project ClimXtreme to combine different methods for a comprehensive assessment of a selected case study, the 29 June 2017 HPE in the metropolitan area of Berlin (Germany), from the meteorological, impacts and climate perspectives, additionally estimating the contribution of climate change to its

extremeness. The main conclusions of the study are:

1. The event occurred under the influence of a mid-tropospheric trough over western Europe (TrW pattern), with two principal short-wave surface lows located east of the British Isles (Rasmund) and over western Poland (Rasmund II) between 29 and 30 June. Rasmund II induced a southwesterly flow bringing moist and warm air ($\theta_e^{850}$ = 306 K, IWV = 42 mm), accompanied by moderate ML-CAPE over the Berlin area ($250\,\mathrm{J\,kg^{-1}}$). Low-level wind convergence along

the German-Polish border triggered several thousand convective cells starting 5 UTC (29 June) that displaced slowly due to the weak tropospheric horizontal winds ($10\,\mathrm{m\,s^{-1}}$ at 500 hPa). Lagrangian backward trajectories determined that the continental region between Poland and northern Italy was the major moisture source feeding the systems with uptakes up to 4.6 mm. Lightning activity was especially active over this area between 05 UTC and 19 UTC (29 June), displacing towards the (northern) Baltic Sea by 23 UTC. Total extreme precipitation amounted locally to 196 mm in 24 h, especially

impacting the Berlin area and the southern limit of the Mecklenburg-Western Pomerania state.

2. The analysis of the impacts showed that the 29 June 2017 case was the costliest event between 2002 and 2017 in the greater Berlin area, with €60 Mill. insured losses in properties due to widespread inundation, traffic disruptions and basement flooding. A set of unique surveys estimating the losses at the household level allowed us to compare the 2017 HPE in Berlin and Brandenburg with previous historical cases in Hersbruck and Lohmar in 2005, Osnabrueck in 2010

and Muenster in 2014. The 2017 HPE stands out in terms of median inundation duration, which was up to 100 h, 4 to 12 times longer than the other surveyed events. While surveyed flood attributes (inundation depth and duration) can be linked to household losses, it is difficult to establish a link between these attributes and the meteorological characteristics



of the event. This is because the impact at the household level is not only caused by the meteorological characteristics, but also by the conditions on the ground.

3. In addition to the impacts perspective, we categorized the event as extreme in terms of frequency of occurrence and severity compared to the climatology. Based on three approaches using Generalized Extreme Value models (Gumbel, standard and duration-dependent), this event showed return periods larger than 100 years for daily precipitation observations, between the greater Berlin area and the southern limit of Mecklenburg-West Pommerania. For higher temporal resolution observations of this event, the return periods were reduced to 10 to 100 years and 1 to 10 years for aggregations

of 8 and 1 hours, respectively. For spatial averages over an area of $11,100\,km^2$, return periods were of 75 years. Furthermore, a precipitation-based index (PSI) classified the 29 June 2017 case as the $29^{th}$ most severe event in the $1951-2021$ climatology. This event was further identified as extreme by Lagrangian cell tracking, classifying the 29 June 2017 case as the second event in the period $2001-2020$ in terms of number of cells triggered, only behind the historical 2002 flooding in Saxony.

4. The attribution experiments for this case study showed that the thermodynamic climate change signal since the preindustrial era caused a small, but significant increase in heavy precipitation for this event. While total precipitation increased by $4\,\%$, the heaviest precipitation rates showed an intensification of over $10\,\%$. Moreover, aerosol sensitivity experiments showed larger cloud mass and increased probability of extreme precipitation rates ($> 150\,\mathrm{mm\,d^{-1}}$), in a polluted scenario compared to the factual situation.

The combined use of impacts, meteorological and climate methods allowed us to relate diverse aspects of heavy precipitation. We found, for the 29 June 2017 event, that large amounts of moisture from continental evaporative sources added to the slow motion of the convective systems to cause one of the most extreme events to date. Furthermore, we related the meteorological extremeness to the impacts for this case, provided the event's large return periods and severity and the reported losses of up to (€60 Mill.). This relationship, however, should not be understood trivially, since precipitation severity is a necessary but not

sufficient condition for high precipitation damages. For the 29 June 2017 event, the state of the ground conditions, i.e. blocking of the sewer system in Berlin and Leegebruch was determinant in inducing the high costs. Finally, the climate attribution experiments demonstrated that a part of the severity of one of the most extreme events of the last 70 years in Germany was attributable to the already-observed changes in the thermodynamical environmental conditions.

Notwithstanding the advantages of multidisciplinary studies, they can suffer from methodological and data inconsistencies,

e.g. when data sets show relevant biases. Whereas discrepancies in the methods cannot be easily overcome, as they are intrinsic to the synergistic approach used in this study, the validation of observations provides a powerful means to reduce uncertainty regarding data biases. Our validation of precipitation data sets in Section 2 showed a high degree of agreement and accuracy between the different products, providing confidence in the conclusions drawn. Among the used products, the accuracy was especially high for REGNIE, followed by RADKLIM.

Also worthy of discussion are the advancements made for the calculation of return periods and fitting of GEV models. While these techniques provide a good estimation of how anomalous an HPE is, our analysis also highlights the large uncertainty of





estimating very large return periods ($> 100$ years) from temporally short databases ($< 70$ years in our case). To overcome this problem, we used station data, which is usually available for longer periods, at different temporal resolutions with a duration-dependent GEV model. Thereby we were able to increase the sample size and shrink the confidence intervals. Other interesting

approaches suggested by previous studies are: considering the climate change signal for $d$-GEV estimation (Ganguli and Coulibaly, 2017), or using spatial models (Ulrich et al., 2020; Berghäuser et al., 2021) that combine information from several stations into one model by using spatial covariates to estimate GEV parameters.

Finally, the multidisciplinary collaboration has created powerful connections that will be exploited in up-coming research. For instance, the authors of this paper will form an expert task force in the framework of ClimXtreme to assess future precipi-

tation events of high interest for the media and the general public, such as the Ahr event in 2021, shortly after their occurrence.

*Code availability.* The WRF model source code can be obtained from https://github.com/wrf-model/WRF/archive/refs/tags/v4.2.1.tar.gz. The COSMO-CLM model is accessible to members of the Climate Limited-area Modeling Community, and access is granted upon request. Parts of the model documentation are freely available at https://doi.org/10.1127/0941-2948/2008/0309 (Rockel et al., 2008)

*Data availability.* REGNIE, RADKLIM, German precipitation station and CatRaRe data from the DWD used in this paper are freely avail-

able for research under the Open Data Portal (DWD, n.d.-a). Lightning EUCLID data are not freely available but can be requested from the Blitz-Informationsdienst von Siemens (https://new.siemens.com/de/de/produkte/services/blids.html). ERA5 data are available via the Copernicus Climate Change Service (C3S; https://climate.copernicus.eu/; last access: 01 January 2022)(Hersbach et al., 2018). ECMWF analysis data can be obtained from https://apps.ecmwf.int/archive-catalogue/?type=an&class=od&stream=oper&expver=1 (last access: 24 August 2021) (ECMWF, 2021). The user's affiliation needs to belong to an ECMWF member state. The WRF model simulation data can be made

available by TS upon request. The COSMO-CLM simulations are being deposited in an open-access repository at the World Data Centre for Climate (WDCC) under the permanent link https://cera-www.dkrz.de/WDCC/ui/cerasearch/entry?acronym=DKRZ_LTA_1152_ds000301. The survey data of the Lohmar (2005), Hersbruck (2005) and Osnabrück (2010) events are available via the German flood loss database HOWAS21 at https://doi.org/10.1594/GFZ.SDDB.HOWAS21 (German Research Centre for Geosciences GFZ, 2022). The data of the Münster (2014) event will be added to HOWAS21 by the end of 2023 and can currently be provided upon request. The Leegebruch (2017) and

Berlin (2017) surveys are available upon request. ICON simulations (aerosol attribution) are archived in long-term storage at DKRZ and as redundant copy at Leipzig University, please send data requests to RC.

*Author contributions.* All authors collaborated and contributed to drafting, reviewing and editing the paper. In particular, ACA and JQ coordinated the effort and wrote the abstract and conclusions of the manuscript; KB, LD, MH, EM, SM, JQ, LS, TS and JST wrote the introduction; SS, DN and FK provided support for data management and precipitation dataset validation; AK performed the Lagrangian

cyclone analysis; MA, SM, TS and VW analysed the synoptic and mesoscale conditions; FR and SP performed moisture backward trajectory analysis; GA, LD, MH, HK, LS and AHT provided the impact analysis and household surveys; JST, SM and FF implemented the generalized extreme value models and estimated the return periods; HF, EELE and ACA implemented the precipitation severity index; KB implemented



the Lagrangian cell tracking algorithm; EM contributed the conditional event attribution experiment; JQ and RC performed the aerosol sensitivity experiments.

*Competing interests.* The authors declare that they have no competing interests.

*Acknowledgements.* This study is the outcome of collaborative work in the "ClimXtreme" project funded by the German Ministry of Education and Research (Bundesministerium für Bildung und Forschung, BMBF) in its strategy "Research for Sustainability" (FONA). The following subprojects have been funded by ClimXtreme. ACA and HF (project SEVERE) and SM and MA (project VarCLuST) sincerely thank the BMBF for funding the two projects (grant number 01LP1901A). TS is funded by the project LAFEP (grant number 01LP1902D);
EM is funded by the project XPreCCC (grant number 01LP1902H). RC and JQ gratefully acknowledge PATTERA (FKZ 01LP1902C). KB is funded by the project DCUA (grant number 01LP1905A). AK sincerely thanks the project B3.6 LAMCoX (grant number 01LP1902F) for their funding and support. HK acknowledges the project FLOOD (01LP1903E). FR and SP are funded by the project MExRain (grant number 01LP1901C). The contribution regarding IDF curves was sponsored by BMBF and the project IDF-AF (grant number 01LP1902H). JST is funded through the project AXE-G (grant number 01LP1902B). EELE is funded by the project CoSoX (grant number 01LP1904C).
GA and LD are funded by the project CARLOFFF (BMBF, grant number 01LP1903B), LS by the DFG (GRK 2043/2). DN and SS are funded by CoDaX (BMBF, grant number 01LP1904A). This work used computing resources of the German Climate Computing Center (DKRZ) granted by its Scientific Steering Committee (WLA) under project IDs bb1152 and bm1159. We thank the CLM-Community (https://wiki.coast.hereon.de/clmcom) for maintaining and providing the COSMO-CLM regional climate model. ECMWF is acknowledged for providing the operational IFS analysis data on model levels and for providing the ERA5 data. The ERA5 reanalysis data used in this study
contain modified Copernicus Climate Change Service Information 2020. Neither The European Commission nor ECMWF is responsible for any use that may be made of the Copernicus Information or the data it contains. DWD is acknowledged for providing the RADKLIM and REGNIE data. The WRF simulation was performed on the HPE Apollo Hawk national supercomputer at the High Performance Computing Center Stuttgart (HLRS) under the grant number ipm12835. The survey campaigns on extreme precipitation impacts were conducted and supported by the following projects/parties: in Lohmar and Hersbruck by the project "URBAS — Urban flash floods" (BMBF; 0330701C),
in Osnabrueck by the University of Potsdam, the German Research Centre for Geosciences GFZ and the Deutsche Rueckversicherung AG, in Muenster and Greven by the project "EVUS — Real-time prediction of pluvial floods and induced water contamination in urban areas" (BMBF, 03G0846B), in Leegebruch 2017 by the project "ExTrass - Urban resilience against extreme weather events–typologies and transfer of adaptation strategies in small metropolises and medium-sized cities" (BMBF; 01LR1709A1) and in Berlin by the project "NatRiskChange - Natural hazards and risk in a changing world" (DFG; GRK 2043/2). MA and SM thank the Blitz-Informationsdienst von Siemens (BLIDS; namely Stephan Thern) for providing the lightning data. KB gratefully acknowledges the provision of computing time through the Center for
Information Services and HPC (ZIH) at TU Dresden.



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
