# Peer review of "Meteorological, Impact and Climate Perspectives of the 29 June 2017 Heavy Precipitation Event in the Berlin Metropolitan Area"

_Natural Hazards and Earth System Sciences, 2022_

## Referee Comment (RC1)

Formal Review for Natural Hazards and Earth System Sciences

Manuscript identification number: **nhess-2022-96**

Title: Meteorological, Impact and Climate perspectives of the 29 June 2017 Heavy Precipitation Event in the Berlin Metropolitan Area

Authors:

Alberto Caldas-Alvarez, Markus Augenstein, Georgy Ayzel, Klemens Barfus, Ribu Cherian,
Lisa Dillenardt, Felix Fauer, Hendrik Feldmann, Maik Heistermann, Alexia Karwat, Frank Kaspar, Heidi Kreibich, Etor Emanuel Lucio-Eceiza, Edmund P. Meredith, Susanna Mohr, Deborah Niermann, Stephan Pfahl, Florian Ruff, Henning W. Rust, Lukas Schoppa, Thomas Schwitalla, Stella Steidl, Annegret H. Thieken, Jordis S. Tradowsky, Volker Wulfmeyer, and Johannes Quaas

**Recommendation: The manuscript may become acceptable after minor revision**

**General Comments:**

The authors investigate a heavy precipitation event in meteorological, climatological and economical sense. It is shown that the chosen case was indeed considered extreme and had high impact therefore its analysis can be relevant for a wider audience and it also fits well into the scope of NHESS. The authors also included an experiment showing the effect of our warming climate on such extreme precipitation event, which help to put the case into perspective. The authors use valid methods to draw their conclusions. The paper consists many different aspects of the case, which have valuable information but also makes it a bit fragmented. I urge the authors to reconsider, if every part is really needed to support their conclusions and to try to shape it to be more coherent. Now it feels like different short analyses listed together.

**Major Comments:**

Section 2: I suggest this section to be restructured, even spitted into two. I would consider putting subsection 2.2 into the results section and mentioning COSMO-REA6 at the reanalysis section. In the 2.3 subsection it is not well explained why the authors used three different

models. What is the added value in using so many models, could it not be done via one model? Also, the analysis done on WRF simulation could it not be done on the COSMO-CLM simulation, or the aerosol simulations with either COSMO or WRF? I also suggest to separate better the models description from the actual experiment set ups.

Section 3: This section should also be unified and simplified. I would suggest to move the methodology description in subsection 3.2 into the Method section and subsection 3.3 into 3.1. Subsection 3.4 consist some paragraphs which are describing the presented Table 1 and Figure 8, which feels unnecessary to me, I would suggest the authors to list more conclusions here. Subsection 3.5.2 also consists some methodology description, which should be moved to the Method section (PSI, cell tracking). Also it is not quite clear what is the contribution of the aerosol experiment to this case study, why is it a valuable part of the analysis.

**Specific comments:**

Page 2 Line 22 (last sentence) A bit vague.
Page 3 Line 3: I would not call Vb cyclones small-scale disturbances
Page 9 Line 217: How do you know that the spin-up was sufficient?
Page 12 Line 293: I would suggest to use "29 June 2017 HPE" instead just "29 June 2017" (not just here, but in the whole paper)
Page 12 Line 300: originated in the night from 28 to 29, would be clearer.
Page 22 Line 483: (Tab. 2, number 6 in Fig. 11)
Page 24 Line 518: "which exceeds the increase in precipitable water we find", in where?
Page 26 Line 551: Which experiments fits better the observations? What are the conclusion related to this event?
Page 26 Line 567: "between Poland and northern Italy" strange wording
Page 26 Line 568: What does "this area" defines here? Not clear to me.
Page 28 Line 615: Is this a plane for further work?

Figure 7 Why are there so little lightning from the area of Berlin, where the precipitation has its peak?

---

## Author Comment (AC1)

**Author reply to RC1 (nhess-2022-96)**

**General Comments**

*The paper consists many different aspects of the case, which have valuable information but also makes it a bit fragmented. I urge the authors to reconsider, if every part is really needed to support their conclusions and to try to shape it to be more coherent. Now it feels like different short analyses listed together.*

We thank the reviewer for the valuable comments and suggestions to improve the manuscript. We acknowledge that the previous version of the manuscript needed more coherence and that some parts were too large. This was also pointed out in the second revision.

This has been a consequence of our team effort where different working groups have obtained the different results presented. To overcome this issue, we have restructured and reduced the text, following the reviewers' guidelines. The main changes are summarized as follows : a) Section 2 has been divided into two sections presenting the data and methods separately; b) all methodologies are explained in Sect. 3 alleviating the text in the results section; c) we have put together all information concerning the case description, i.e. meteorology, impacts and index analysis; d) the most relevant outcomes of our work, e.g., the moisture source analysis, attribution experiments, etc. have been highlighted being given their own subsection.

We expect these changes to have improved the readability of the text and its structure making it easier for the potential readers to access the relevant results of our work.

**Major Comments**

*Section 2: I suggest this section to be restructured, even spitted into two*

This has been implemented. We divided Section 2 into two new separate sections, "Data" and "Methods"

*I would consider putting subsection 2.2 into the results section and mentioning COSMO-REA6 at the reanalysis section.*

Also implemented (see new version of the manuscript).

*In the 2.3 subsection it is not well explained why the authors used three different models. What is the added value in using so many models, could it not be done via one model? Also, the analysis done on WRF simulation could it not be done on the COSMO-CLM simulation, or the aerosol simulations with either COSMO or WRF?*

This question raised by the reviewer needs explaining. For some purposes of the different analyses, as the reviewer points out, the same model could have been used, e.g., ICON could have been used for the aerosol experiments as well as the water vapour analyses in Fig. 5. We are conscious that this affects the papers readability and length that could have been reduced with the use of just one model.

We planned this research to be able to profit from the different expertise in such a large collaboration. The participating research groups had implemented their analyses and methods on numerical models best known to them or used in their respective institutions. Therefore, different models were used for different experiments. This aspect is intrinsic to our collaboration and the approach of our paper which is somehow unavoidable if we are to combine so many different methods.

On another note, we believe our results to be model independent and we cross-validated our model simulations before starting our analyses. All models showed the same dynamical situation concerning the synoptic evolution, the location of precipitation, the moisture transport and convective activity. Therefore, we are confident that the use of different models does not hamper our drawing of conclusions with the added value of profiting from different analytical methods.

We believe this aspect is worth mentioning, and the following remark is added in the conclusions

*"An example of this, is the use of different numerical models. We decided to use different models to profit from the techniques best known to the different working groups. Moreover, we are confident that our results are model independent since all models showed a similar dynamical evolution of the event. "*

*I also suggest to separate better the models description from the actual experiment set ups.*

This has been implemented. We have split the model description from the experiments' description.

*Section 3: I would suggest to move the methodology description in subsection 3.2 into the Method section and subsection 3.3 into 3.1.*

Implemented. Besides, Sect. 3.4 "Impacts during the 29 June 2017 case" has also moved into Sect 3.1 to provide the description of the case in just one section, now named "Event analysis and climate context.

*Subsection 3.4 consist some paragraphs which are describing the presented Table 1 and Figure 8, which feels unnecessary to me, I would suggest the authors to list more conclusions here.*

We have rewritten former Subsection 3.4 (now contained in Sect. 4.1 "Case analysis and climate context") to convey the conclusions more than a description of the charts.

*Subsection 3.5.2 also consists some methodology description, which should be moved to the Method section (PSI, cell tracking)*

Implemented

*What is the contribution of the aerosol experiment to this case study, why is it a valuable part of the analysis.*

We believe the aerosol experiments to be a relevant part of this analysis since they show another potential impact of anthropogenic activities on heavy precipitation, in addition to global warming. The impact of aerosol on heavy precipitation is of relevance to the 29 June 2017 HPE since it occurred in an urban area which tend to suffer more frequently from pollution.

**Specific Comments**

*Page 2 Line 22 (last sentence) A bit vague.*

It has been rephrased as:

*"For instance, the link between the unique meteorological conditions of this case and its very large return periods or the extent to which it is attributable to already-observed anthropogenic climate change."*

*Page 3 Line 3: I would not call Vb cyclones small-scale disturbances*

We totally agree with the reviewer. We believe the "small-scale disturbances" part of the sentence was inherited from a previous version of the manuscript. It has been corrected.

*Page 9 Line 217: How do you know that the spin-up was sufficient?*

In this sentence, "spin-up" was supposed to refer to the spinning-up of small-scale (convective) precipitation features in the convection-permitting (0.025°) model. For example, in a downscaling from 15 km to 3 km (we have 12 km to 2.8 km) over the central United States, Wang and Skamarock (2016; Fig. 2b) found a value of about 6 - 12 h for the spin-up of small-scale convective precipitation features. In our convection-permitting simulations, the analysis period begins 8 hours after the model is initialised from its 0.11° parent model.

Figure 1 illustrates the spinning-up of small-scale precipitation features in our 0.025° simulations. Here we see the temporal evolution of the spatial standard deviation of (i) column-integrated cloud graupel, (ii) column-integrated cloud ice, and (iii) column-integrated cloud water. Note that these variables (on model levels) were all included in the initial conditions provided by the 0.11° model. A high standard deviation would represent high small-scale variability of these variables. As can be seen, there is a rapid increase in small-scale variability after initialization, with the period of rapid growth completed before the start of the analysis period. For this reason, we believe that the spin-up prior to the analysis period is sufficient.

[Figure]

*Figure 1. Temporal evolution from initialization for the spatial standard deviation of (i) column-integrated cloud graupel, (ii) column-integrated cloud ice, and (iii) column-integrated cloud water. The curves represent the 0.025° convection-permitting model, which was initialized from its 0.11° parent model at 2300 UTC on 28th June 2017. The analysis period -- marked with a vertical black line -- starts at 0700 UTC on 29th June 2017, eight hours later. The spatial standard deviation is computed over the analysis region shown in Figure S4.*

*Page 12 Line 293: I would suggest to use "29 June 2017 HPE" instead just "29 June 2017" (not just here, but in the whole paper)*

We have accepted the suggested terminology for the whole paper, where applicable.

*Page 12 Line 300: originated in the night from 28 to 29, would be clearer.*

Suggestion accepted.

*Page 22 Line 483: (Tab. 2, number 6 in Fig. 11)*

Suggestion accepted.

*Page 24 Line 518: "which exceeds the increase in precipitable water we find", in where?*

The sentence has been rephrased:

*"which exceeds the 7.5 % increase in precipitable water (time average; not shown) which we find over our eastern-Germany analysis region (Fig. S4)."*

*Page 26 Line 551: Which experiments fits better the observations? What are the conclusion related to this event?*

The probability distribution of CLN shows a better agreement with observations for intensities below 80 mm/d as is it can show larger probabilities of precipitation below that value. Both intensities over 80 mm/d both CLN and POL show a similar behaviour. Regarding the spatial distributions shown in Fig. 13 both CLN and POL showed a similar performance and agreement to observations.

These results indicate the a more polluted scenario over an urban area (POL) is likely to shift the precipitation's PDF towards more extreme values, i.e., larger precipitation intensities and lower probability of moderate rates for this event.

*Page 26 Line 567: "between Poland and northern Italy" strange wording*

Has been rephrased to: *"Lagrangian backward trajectories determined that the continental region between southern Poland and northern Italy was the major moisture source feeding the systems, with uptakes up to 4.6 mm"*

*Page 26 Line 568: What does "this area" defines here? Not clear to me.*

The region in central Europe between Poland and the Alps/Adriatic Sea. The sentence has been changed to: "Lightning activity was especially active over the same region area between 05 UTC and 19 UTC (29 June)."

*Page 28 Line 615: Is this a plane for further work?*

Yes, these are possible ideas for further investigation.

*Figure 7 Why are there so little lightning from the area of Berlin, where the precipitation has its peak?*

This event did not show a strong correlation of heavy rainfall and the location of lightning. As stated by Kuhlman et al., (2009), it has been previously observed that lightning activity can have a significant spatio-temporal variation to its accompanying perils. In this sense, a high spatio-temporal lightning density does not necessarily mean a large amount of precipitation at the same spot. Although statistical correlation between cloud to ground lightning and precipitation is generally expected (e.g. Katsanos et al., 2007, Matsangouras et al., 2016).

In our case, the event is characterized by embedded convection on the upstream side and it is possible that precipitation occurred after lightning activity (after charge separation in the cloud). This can explain the lack of co-location of both perils.

**References**

Kuhlman, K. M., MacGorman, D. R., Biggerstaff, M. I., & Krehbiel, P. R. (2009). Lightning initiation in the anvils of two supercell storms. Geophysical Research Letters, 36(7). doi: 10.1029/2008GL036650

Wong, M., & Skamarock, W. C. (2016). Spectral characteristics of convective-scale precipitation observations and forecasts. Monthly Weather Review, 144(11), 4183-4196.

---

## Author Comment (AC2)

**Author reply to RC2 (nhess-2022-96)**

**Major Comment**

*In my opinion, the most interesting parts of the paper are contained in Sections 3.2 (Lagrangian Moisture source analysis), 3.6.1 (Conditional event attribution) and 3.6.2 (Role of Aerosols). Therefore, I would give more space to these parts, that, in the present version of the paper. Therefore, I would give more space to these parts, that, in the present version of the paper, have a marginal role. Contrarily, I would shorten the other Sections, which basically contain a description of the event. For example, I would include Section 3.3 (Observed lightning activity and accumulated precipitation) and 3.5 (Probability of exceedance and severity) in a new Section 3.1, which would contain the complete description of the event, from the synoptic scale to the impacts at the local scale. This, in my opinion, would improve the readability of the paper. The analyses that go beyond the simple description of the event may be included in subsequent Subsections (or in a new Section 4).*

We thank the reviewer for the valuable comments and suggestions to improve the manuscript. We acknowledge that the previous version of the manuscript needed more coherence and that some parts were too large. This was also pointed out in the second revision.

We have followed the reviewer's advice and the most relevant results have been given a separate subsection (within the results section). Moreover, former Section 2, has been split into two ("2. Data" and "3. Methods") ,and Sections 3, "Meteorological situation", 3.3 "Observed lightning and accumulated precipitation", 3.4 "Impacts during the 29 June 2017 case", and 3.5.2 "Severity" have been merged into one ("4.2 Case analysis and climate context"). Section 3.5.1 "Probability of exceedance" has been given a subsection within the results, as we believe the discussion presented here will also be of high interest to potential readers-

We expect these structural changes to have improved the readability and presentation of our results, as pointed out by the reviewer.

**Minor and technical comments**

*Line 113: lightNings*

Corrected.

*Line 178: is the resolution of ERA5 25 x 25 km2 or 31 x 31 km2, as stated at line 150?*

The resolution is 31 km. This has been corrected in the text.

*Line 230: corresponding approximately TO the peak…*

Corrected.

*Line 480: I do not understand why the 29 June 2017 event is the 29th most severe event in the 1951-2021 period.*

We agree that the writing of this sentence is misleading. The information we wanted to deliver here is that, the 29 June 2017 event showed the 29[th] highest PSI value.

The PSI is an index that assigns values typically between zero and 2, depending on how severe a record of daily precipitation was. The severity is calculated according to three magnitudes, a) grid-point daily

precipitation intensity, b) extent of surface affected by precipitation larger than the percentile-80 of the climatology and 3) the persistence of precipitation over the same grid point.

In this sense, after obtaining a value of the PSI for every day in the period 1951-2021, the 29 June 2017 HPE, had showed the 29th highest value compared to the all other days in the 70 -year climatology.

We have adapted this sentence. Now it reads:

*"The unusual precipitation totals made the 29 June 2017 HPE one of the most extreme event in the climatology of the greater Berlin area. Based on the PSI method (Sect.3.1; Caldas-Alvarez et al., 2021; Piper et al., 2016), the 29 June 2017 event around Berlin was the 29th most severe event in the 1951-2021 period (Fig. 6). This event showed a PSI value (1.71), well above the 99th-percentile of the climatology indicating an extreme event. The PSI quantifies the severity of an event considering grid-point precipitation intensity, surface of affected area and persistence.*

*Line 486: The Ahr flooding in July-2021 (number 7 in Fig. 11). Should be number 9?*

Corrected.

*Line 488: due to the lack OF observations*

Corrected.

*Line 563: Rasmund II induced… Rasmund II or Rasmund?*

It was Rasmund II.

The upper-level large-scale trough favoured a generalized cyclonic circulation over western Europe. However, it was Rasmund II, a surface low of relatively small-scale, the disturbance that deepened the penetration of warm and moist air from the south (Slovenia, northern Italy) up to the Berlin area. This is can be seen in Fig. 3 of the manuscript where the higher values of $\theta_e$ occurred in the warm sector of Rasmund II.

Because Rasmund originated earlier (British Isles), it was given the principal name and Rasmund II (Central Europe) was given the ordinal suffix (II) as it originated several hours later.

We have added labels in Fig.3 to enhance the readability and comprehension of these results in the manuscript.

*Line 565: triggered several thousand convective cells... is "several thousand" correct?*

We corrected this to "over 11.000 convective cells, to be more precise

*Line 568: Lightning activity was especially active…repetition*

Corrected

*Figure 2: what do the vertical lines in panel b represent?*

The vertical lines in figure 2b indicate the standard error (SE) of the computed SEDI, with $SE = \sigma/\sqrt{n}$

*Figure 12: Is this Figure referring only to the area surrounding Berlin, as written in the caption? If yes, what area? Panel b of Figure 1?*

Yes, it is a squared area of 350 km x 350 km, covering the Berlin metropolitan area, reaching the polish border as shown in the following figure.

[Figure]

*Figure 1. Isolated convective cells identified through the cell tracking algorithm over the Berlin metropolitan area. A surface of 350 km x 350 km is selected for identifying the cells*

We have adapted the caption of Fig. 12 to better describe the extent of this area.

*Moreover, can you give more information on how data are divided into the two groups?*

The objective weather type classification of Bisolli and Dittmann (2001) has been derived for the focus region by centering the analysis domain over Berlin (the overall analysis region is much bigger covering huge parts of Central Europe). Input data have been ERA5 data for 12 UTC (except for the humidity classification with daily data). We used the same levels and the same reference period as described in Bisolli and Dittmann (2001). So, we end up with one weather type for each day in the period 2001 to 2020 (the period of the radar data). In the plot we distinguish days which belong to the weather type 'no prevailing wind direction, cyclonic circulation in 950 and 500 hPa and above-average humidity content of the troposphere - XXCCW and days belonging to another weather type (the exact type is known but not plotted here).

**References**

Bisolli, P. and Dittmann, E.: The objective weather type classification of the German Weather Service and its possibilities of application to environmental and meteorological investigations, Meteorol. Z., pp. 253–260, https://doi.org/10.1127/0941-2948/2001/0010-0253, 200

---

## Author Response (AR2)

Referee #1
The Authors answered satisfactorily my previous comments.
*We thank the reviewer for their continued guidance improving the manuscript.*

Technical revisions:
Line 185: Nopn-hydrostatic.
*corrected*

Lines 190-191: These sentences are not clear.
*We simplified to read: "This section introduces the methods used in the different aspects of the analysis."*

Line 287: delete "at".
*done*

Line 329: there is something wrong with this sentence.
*Corrected by replacing "with" by "while" and "by" by "for".*

Line 359: I still do not understand, from Fig. 6, why the episode studied is the 29th (from the figure it seems to me it is the 1$^{st}$)
*Perhaps there is a mistake in interpretation possible. In order to make this more clear, we now revise the Caption of Fig. 6 to clarify this is for the past 70 years so the distinction between two days is really difficult.*

Line 529: comma and not full stop before "in order".
*corrected.*

*Reviewer #2*
The manuscript has a clear structure and represents the results understandably with valid conclusions.
I have only minor comments.
*We thank the reviewer for once again going through the manuscript and responses carefully.*

1)Page11 Line313: "For this reason ERA5 will not be used hereafter to study heavy precipitation fields. Its use will be restricted to other relevant large-scale atmospheric fields such as water vapour (Fig. 9)" I would suggest to finish this sentence with a reason why the authors believe ERA5 large scale fields are still a reliable source despite the short comings in precipitation.
*This is a good suggestion, we add the reason: "...that are well constrained by the data assimilation."*

2)Page16 Line 379: Impacts and monetary losses, isn't here a subsection sign missing, so 4.2?
*The reviewer is right, it is much better to have this a numbered subsection.*

3)Page16 Line 381: Is it possible to updated the period until 2022?
*Very good point! Indeed we can do this, based on the available data, and reformulate accordingly (sentence simplified to "...most damaging extreme event in Berlin and Brandenburg since 2002."*